# Intriguing role of water in protein-ligand binding studied by neutron crystallography on trypsin complexes

Johannes Schiebel[1,2], Roberto Gaspari[2], Tobias Wulsdorf[1], Khang Ngo[1], Christian Sohn[1], Tobias E. Schrader [3], Andrea Cavalli[2], Andreas Ostermann [4], Andreas Heine[1] & Gerhard Klebe[1]

Hydrogen bonds are key interactions determining protein-ligand binding affinity and therefore fundamental to any biological process. Unfortunately, explicit structural information about hydrogen positions and thus H-bonds in protein-ligand complexes is extremely rare and similarly the important role of water during binding remains poorly understood. Here, we report on neutron structures of trypsin determined at very high resolutions $\leq$1.5 Å in uncomplexed and inhibited state complemented by X-ray and thermodynamic data and computer simulations. Our structures show the precise geometry of H-bonds between protein and the inhibitors *N*-amidinopiperidine and benzamidine along with the dynamics of the residual solvation pattern. Prior to binding, the ligand-free binding pocket is occupied by water molecules characterized by a paucity of H-bonds and high mobility resulting in an imperfect hydration of the critical residue Asp189. This phenomenon likely constitutes a key factor fueling ligand binding via water displacement and helps improving our current view on water influencing protein–ligand recognition.

---

[1] Institut für Pharmazeutische Chemie, Philipps-Universität Marburg, Marbacher Weg 6, 35032 Marburg, Germany. [2] Computational Sciences, Istituto Italiano di Tecnologia, Via Morego 30, 16163 Genova, Italy. [3] Jülich Centre for Neutron Science at Heinz Maier-Leibnitz Zentrum, Forschungszentrum Jülich, Lichtenbergstraße 1, 85748 Garching, Germany. [4] Heinz Maier-Leibnitz Zentrum, Technische Universität München, Lichtenbergstraße 1, 85748 Garching, Germany. Correspondence and requests for materials should be addressed to G.K. (email: klebe@mailer.uni-marburg.de)

Bio-macromolecules evolved their function in an aqueous milieu[1]. In the bulk phase, water constitutes a dynamic network of individual molecules interlinked by H-bonds, which is modulated in the contact region with solvated molecules. This modulation depends on the shape and physicochemical properties of the solute. In vivo, proteins and their ligands are surrounded by water molecules that need to be shed, entrapped, rearranged, and modulated in their dynamic properties when the water-exposed surfaces of the proteins or ligands change for instance upon formation of assemblies such as enzyme-substrate or protein-inhibitor complexes[2]. Thus, the solvent water necessarily has a major impact on the binding event and energetics. The role of water molecules during ligand recognition is believed to be multifactorial and highly complex explaining why binding affinity is often difficult to predict even when structural information is available[2,3].

Based on their unique properties, water molecules fulfill multiple functions in protein–ligand complexes, e.g. by mediating hydrogen bonds between interaction partners due to their dual ability to act either as donor or acceptor[3]. The displacement of water molecules from protein-binding pockets during ligand association also impacts affinity and governs enthalpy/entropy partitioning according to properties of the individual water molecules in the pocket compared to those in the bulk phase[4]. In particular, it has been proposed that the expulsion of thermodynamically unfavorable water molecules during complex formation enhances the ligand's affinity[5]. Although a sound experimental basis supporting this hypothesis has not yet been established, the computational estimation of the energetic properties of individual binding-site water molecules via molecular mechanics methods emerged as a valuable tool to improve the prediction of binding affinity[6]. Additionally, it should be noted that not only the protein but also the ligand itself needs to be, at least partially, desolvated and thereby influences binding.

Although the importance of water molecules during ligand binding is generally accepted and some key concepts have been proposed and computationally studied, experimental and in particular structural data deciphering how water molecules act exactly during protein-ligand complex formation unfortunately is rare[7–10]. In part, this lack of knowledge arises from the fact that macromolecular X-ray crystallography can hardly detect hydrogen atoms reliably unless ultra-high resolution data had been collected, which may allow detection of the structurally most defined H-atoms. Therefore, particularly the orientational and rotational states of the three atoms forming a water molecule remain unresolved at resolutions typically achieved by this technique[11]. In contrast, neutron crystallography delivers structures that contain substantial information concerning hydrogen atoms but this technique is challenging to apply as obvious from the very low number of available neutron structures in the PDB (0.091% of all entries as of the end of July 2017)[12]. To some degree this challenge arises from the facts that very large single crystals are required and access to neutron sources is still limited. Current improvements, such as the usage of fully deuterated protein material for crystallization, and the construction of next-generation neutron beamlines hold the promise to enable more neutron-diffraction studies in the future. Nevertheless, even if a neutron structure can be determined successfully with the usually achieved resolution of ~2.0 Å, it remains difficult to unambiguously elucidate the orientation of many water molecules due to their mobility or disorder[13,14]. The situation improves significantly once the resolution gets better than 2.0 Å with the orientation of many waters discernible at 1.5 Å or better[13]. Such high resolution, however, is extremely rare[14].

To substantially improve our understanding of ligand binding events, it would not even be sufficient to determine the structure of a protein-ligand complex by neutron diffraction at the required high resolution, but it would also be key to provide a high-quality neutron structure of the uncomplexed state of the protein, which describes the binding pocket in its hydration state prior to ligand association. In an attempt to accomplish this goal, we chose the digestion enzyme trypsin for our studies as a representative of the large class of serine proteases. This protein family includes many disease-related proteins targeted in drug discovery campaigns[15]. Individual water molecules located in the deep $S_1$ pocket of the trypsin-like serine proteases are known to drastically alter the ligand's free enthalpy of binding depending on changes induced upon complex formation[6]. For instance, the expulsion of a single water molecule located above Tyr228 from the $S_1$ pocket of the blood coagulation factors IIa (thrombin) and Xa results in an affinity boost. This finding paved the way for the development of orally bioavailable anticoagulants lacking the originally used benzamidine anchor, long time believed to be essential for binding. Despite the success of this and similar strategies, experimental studies accurately characterizing and explaining the interplay between protein, ligand and water are still rare. Here, we determined neutron structures of bovine trypsin at exceptional resolutions better than 1.5 Å in the uncomplexed state and subsequently in the bound state with the two chemically related ligands N-amidinopiperidine and benzamidine. These data have been augmented by ultra-high resolution X-ray structures better than 1.0 Å determined at room and cryogenic temperature as well as by biochemical and biophysical binding data. The experimental studies were complemented by an in-depth computational characterization of both ligand-binding events using metadynamics and QM-based approaches, which provide a detailed structural and dynamic view of protein-ligand complex formation.

## Results

**Ligand selectivity trypsin/thrombin.** Serine proteases such as trypsin and thrombin specifically recognize substrates comprising basic residues prior to the peptide bond cleavage. The basic $P_1$ residue interacts with Asp189 at the bottom of the $S_1$ specificity pocket[16]. While thrombin nearly exclusively accepts arginine side chains, trypsin effectively hydrolyzes both, arginine and lysine-based substrates. Benzamidine is a well-known alkaline inhibitor that blocks proteolytic function of both and interacts with Asp189 in an arginine-analogous manner[15]. In the search for thrombin-selective inhibitors, Hilpert et al. identified N-amidinopiperidine as a promising lead that served to anchor inhibitors in the $S_1$ pocket of thrombin providing a significant selectivity advantage over trypsin binding. Subsequently, the N-amidinopiperidine head group was evolved to the drug napsagatran, which had been under clinical investigation as anticoagulant[16]. We selected both compounds for their frequent incorporation in potent serine protease inhibitors mimicking arginine substrates.

**Thermodynamic binding signature.** First, we characterized the binding process by means of a fluorescence-based inhibition assay and isothermal titration calorimetry (ITC) following a direct as well as a displacement titration protocol. Benzamidine inhibits trypsin about 10-fold stronger than N-amidinopiperidine while the situation is inverted for thrombin, which favors binding of N-amidinopiperidine by a factor of 2 over trypsin (Table 1)[16]. These inhibitory properties are confirmed by the D-Phe-Pro-analogs of benzamidine and N-amidinopiperidine, which also suggest strong preference for trypsin by benzamidine-like head groups (Supplementary Table 1). This observation can be attributed to a clear enthalpic advantage observed for benzamidine over N-amidinopiperidine binding (Supplementary Tables 2 and 3).

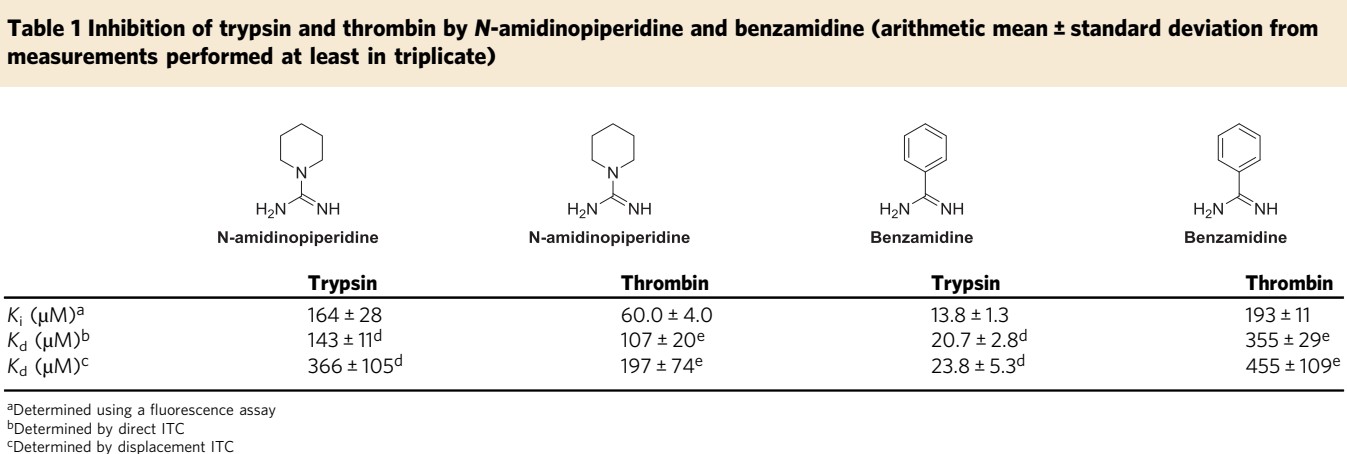

**Table 1 Inhibition of trypsin and thrombin by *N*-amidinopiperidine and benzamidine (arithmetic mean ± standard deviation from measurements performed at least in triplicate)**

| | N-amidinopiperidine | N-amidinopiperidine | Benzamidine | Benzamidine |
| --- | --- | --- | --- | --- |
| | **Trypsin** | **Thrombin** | **Trypsin** | **Thrombin** |
| $K_i$ (µM)[a] | 164 ± 28 | 60.0 ± 4.0 | 13.8 ± 1.3 | 193 ± 11 |
| $K_d$ (µM)[b] | 143 ± 11[d] | 107 ± 20[e] | 20.7 ± 2.8[d] | 355 ± 29[e] |
| $K_d$ (µM)[c] | 366 ± 105[d] | 197 ± 74[e] | 23.8 ± 5.3[d] | 455 ± 109[e] |

[a]Determined using a fluorescence assay
[b]Determined by direct ITC
[c]Determined by displacement ITC
[d]Determined in HEPES buffer
[e]Taken from ref. [64]

Next, we attempted to describe the structural states prior and subsequent to ligand binding as precisely as possible and managed to collect neutron diffraction data of trypsin in its uncomplexed as well as benzamidine and *N*-amidinopiperidine bound state to very high resolutions of 1.45–1.50 Å (Supplementary Table 4). This was augmented by ultra-high resolution X-ray data at cryogenic (0.86–1.01 Å; Supplementary Table 5) and ambient temperature (0.93–0.99 Å; Supplementary Table 6). To optimally exploit these data, we refined the structures either against neutron and X-ray data separately, and additionally we determined joint X-ray/neutron (XN) structures (Supplementary Table 7). Combining the data collected at the same temperature greatly facilitated identification of the orientation of individual water molecules and of certain protein functional groups such as hydroxyl moieties[17].

**Hydration pattern of uncomplexed trypsin.** Prior to ligand association, the uncomplexed protein-binding pocket exhibits a specific hydration pattern, which changes upon ligand binding thereby influencing affinity and the thermodynamic binding signature. The XN structure of uncomplexed trypsin unravels how water molecules surround Asp189 in detail. They can be subdivided into two groups: Waters W1–W3 are located within the $S_1$ pocket while W4–W7 flank Asp189 from behind and form a kind of water reservoir more deeply buried within the protein (Fig. 1a). Such waters found in internal pockets typically form at least three hydrogen bonds[18]. Indeed, the four reservoir waters W4–W7 engage on average in about three H-bonds (Fig. 1b). In contrast, the other $S_1$ pocket water molecules are characterized by a paucity of interactions with an average of only two H-bonds. Based on these findings and an equation derived by Yu et al.[19], we estimate that the $S_1$ pocket waters are on average by about 10 kJ mol$^{-1}$ less stable than the reservoir waters. The reduced number of H-bonds of the $S_1$ pocket waters is reflected in a considerably increased disorder evident from the nuclear and electron density of our XN structure (Fig. 1a). W1 and W3 could be refined in two rotational states, which is consistent with the short-time rotational behavior found in a corresponding molecular dynamics (MD) simulation of the unbound state (see Supplementary Note 1), while the intercalating water W2 seems to switch between two positions depending on the assumed W1 and W3 orientations (Fig. 1b, one configuration of the water structure is highlighted in cyan, the other in pink). At room temperature, both $S_1$ water configurations are approximately equally populated. When lowering the temperature to 100 K during X-ray diffraction experiments, W2 was found to be rather H-bonded to

W1 than to W3 (occupancies of 75 and 25% in our structure with PDB-code 5MNE). To analyze whether this temperature effect is relevant, we collected three additional X-ray diffraction data sets from non-deuterated uncomplexed trypsin crystals each at 295 and 100 K. All structures were equally processed via an automated refinement pipeline[20], which places one W2 molecule into the center of the corresponding electron density feature. The resulting structures confirmed that the water pattern in the $S_1$ pocket differs at the two temperatures with W2 being much closer (by 0.4 Å) to W1 at 100 K (Fig. 1c). These findings suggest the existence of a double well on the free energy landscape. Using the refined W2 occupancies at room and cryogenic temperature (PDB-codes 5MOP vs. 5MNE), we estimated according to a Boltzmann distribution that the energy difference between both states amounts to ~0.8 kJ mol$^{-1}$.

Such an in-depth characterization of the solvation pattern of the uncomplexed enzyme is crucial for understanding protein-ligand binding, as the difference in the water pattern of uncomplexed and ligand-bound state suggests how many and which H-bonds must be broken upon complex formation. The on average lower number of H-bonds formed by the water molecules in the $S_1$ pocket in the uncomplexed state are likely key to fuel binding of inhibitors and substrates. In addition, Asp189 seems imperfectly hydrated. In particular, one carboxylate oxygen lone pair pointing toward the $S_1$ pocket is not saturated by any H-bond donor provided by a water molecule (Fig. 1b). We hypothesize that this situation is caused by the shape and orientation of the $S_1$ pocket. Specifically, the methylene group of Gly226 does not allow W2 to approach the non-saturated Asp189 lone pair head-on although this would be the preferred geometry for an H-bond donor. This can be seen from the perfect solvation of the more freely accessible Asp71 (Fig. 1d–f)[21]. The imperfect hydration of Asp189 is supposedly a prerequisite for potent ligand binding because desolvation of the charged carboxylate in the deep S1 pocket would otherwise be energetically too costly. This is even more important since the preferred trypsin substrates and most inhibitors, including those investigated here, contain positively charged groups that also need to be desolvated upon uptake into the binding pocket.

The dynamical behavior of the water molecules in the $S_1$ pocket was studied by MD simulation of uncomplexed trypsin. It was found that the mean residence time of water molecule W1 is much shorter than for W2 or W3 (10.6 ps for W1 and 87.6 ps for W2/W3). In order to elucidate the impact of the protein environment on the solvation dynamics next to Asp189 (solvent sites W2/W3) and Tyr228 (solvent site W1), we normalized all

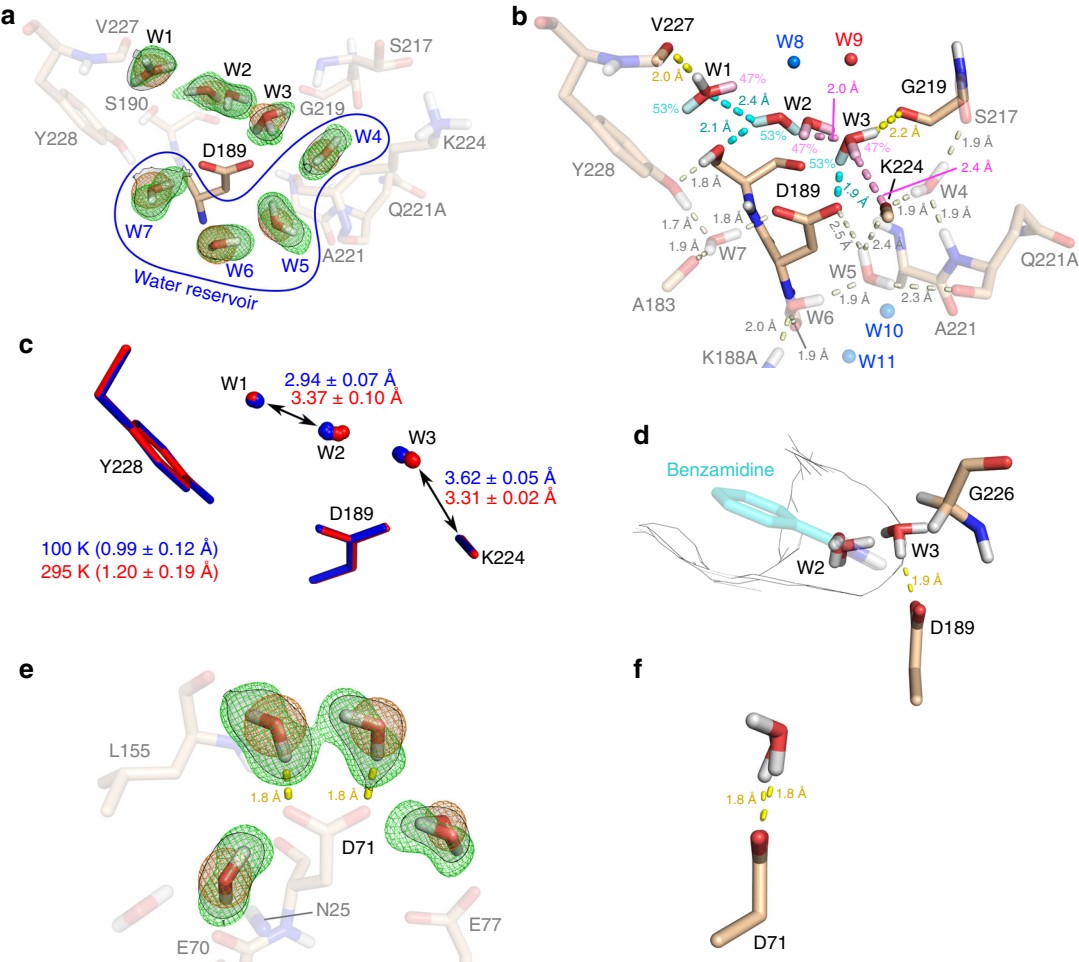

**Fig. 1** Solvation of the uncomplexed $S_1$ pocket. Protein residues from the XN structure of trypsin in its uncomplexed state are depicted as light brown sticks. **a** Water network surrounding Asp189. X-ray and neutron $2mF_o$-$DF_c$ maps at 1.5 and 1.8σ are shown as orange meshes and transparent gray surfaces surrounded by black outlines, respectively, while green meshes represent the neutron $mF_o$-$DF_c$ omit map at 4σ obtained after removal of deuterium atoms from all depicted water molecules. The water reservoir next to the $S_1$ pocket made up by waters W4–W7 is highlighted in blue. **b** Interactions between waters and amino acids within the $S_1$ pocket of trypsin. Interactions are represented by dashed lines labeled with deuterium•••acceptor distances. For waters W1, W2, and W3 two alternative conformational states could be modeled in the uncomplexed XN structure. The two states are colored differently with the hydrogens of the higher populated state (53%) including their interactions in cyan and of the lower populated state (47%) in pink. For water W9 only the oxygen was visible in the XN structure. In contrast, waters W8, W10, and W11 have not been modeled in the room-temperature XN structure at all but were clearly visible at cryogenic temperature and are shown from a superimposition of the X-ray cryo onto the XN structure as blue spheres. **c** Temperature-dependent change in the Asp189 solvation pattern. Four individual X-ray structures of uncomplexed trypsin at cryogenic temperature that were determined via an automated refinement pipeline are shown in blue[20]. The data for these structures have been collected from one deuterated and three non-deuterated crystals. Similarly, four room-temperature structures are depicted in red. All structures have been superimposed. Mean resolutions and distances (±standard deviation) are indicated for both temperature settings. **d** Orientation of Asp189 relative to the $S_1$ pocket. The $S_1$ pocket is highlighted as Connolly surface outline. **e**, **f** Hydration of Asp71. Maps are shown as described for **a**

residence times relative to those obtained for entirely solvent-exposed aspartate and tyrosine residues. Normalized residence times of W2/W3 were found to be longer than those of W1 indicating a stronger influence of the environment on W2/W3 compared to W1 (normalized residence times are 14.0 and 5.73 for W2/W3 and W1, respectively; for details see Supporting Results section). It is known that charged residues like Asp189 tend to have slower solvent exchange rates than polar or apolar groups[22]. However, it is unclear why these are still slower when applying normalization as described above. It can be reasoned that the shape and the electrostatic properties of the binding pocket have a large impact on the unbinding dynamics of W2/W3, thus enhancing interactions arising from electrostatic contributions and hydrogen bonding abilities of Asp189. A similar conclusion was also drawn in a work by Makarov et al.[23],

who investigated the residence times of water molecules in myoglobin. Moreover, not only the protein environment modulates the characteristics of W1 and W2/W3, but also the rest of the water network in the S1 pocket.

Upon ligand or substrate binding to trypsin, the neatly assembled water network, which likely plays a functional role during binding processes, will be disrupted. The pronounced difference in residence time between the neighboring solvent sites W1 and W2/W3, as described above, can be understood as an important feature in the desolvation mechanism of the S1 pocket. Especially the fast exchange rate of the W1 solvent site and the corresponding low barrier for water unbinding might represent the main pathway for pocket desolvation during ligand binding.

A further conclusion can be drawn on the role of W3 water molecules, which, as will be seen from our metadynamics

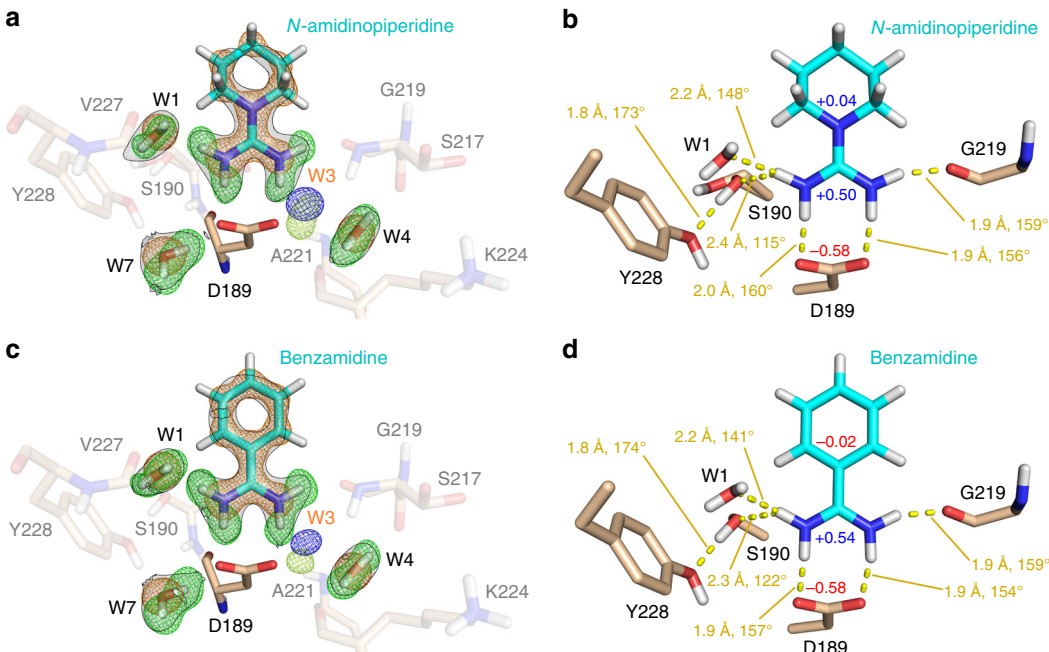

**Fig. 2** Hydrogen-bonding network formed between trypsin, water and basic inhibitors. The XN structures of trypsin in its complexes with *N*-amidinopiperidine and benzamidine are shown with protein residues as light brown and ligands as cyan sticks. **a, c** H atoms in the structures of trypsin complexed with *N*-amidinopiperidine or benzamidine, respectively. X-ray and neutron 2mF$_o$-DF$_c$ maps are shown at the 1.8σ level as orange meshes and transparent gray surfaces surrounded by black outlines, respectively. Corresponding mF$_o$-DF$_c$ maps are depicted at 3σ in blue and light green surrounding the site for water W3, which has not been modeled in the structures due to insufficient occupancy. Moreover, neutron mF$_o$-DF$_c$ omit maps have been generated after removal of all hydrogens from each ligand and all depicted water molecules. These maps are shown at 5σ for ligand and water deuterons in green. **b, d** Contacts between trypsin and *N*-amidinopiperidine or benzamidine, respectively. Individual interactions are represented by yellow dashed lines, which are labeled with the distance between the H-bond donor deuterium and corresponding acceptor atom along with the donor-D•••acceptor angle. Mulliken charges (red) obtained as the mean from a short QM/MM trajectory are depicted for the amidino groups of both ligands, the attached ring atom and the Asp189 carboxylate

simulations, assist in the stabilization of the ligands in a pre-binding state. From the slow dissociation rate of these W3 water molecules, we assume that they impose the lower boundary on the rate at which ligand molecules can actually penetrate into the binding pocket and reach the aforementioned pre-binding state (for details, see Supplementary Note 1 and Supplementary Tables 8–10).

**Binding-poses of benzamidine and *N*-amidinopiperidine.** Similar to the protein's state prior to ligand binding, we analyzed the benzamidine and *N*-amidinopiperidine-bound complexes of trypsin. Both inhibitors bind in very similar fashion to the S$_1$ pocket (Fig. 2). Two amidino group hydrogens interact directly with the carboxylate oxygens of Asp189 while one additional hydrogen is contacting Gly219(C=O). Interestingly, the fourth hydrogen engages in a bifurcated hydrogen bond with W1 and Oγ of Ser190. For these H-bonds, both inhibitors do not differ significantly (Fig. 2b, d). The amidino-carboxylate interactions are enhanced by an electrostatic component. Short QM/MM simulations for both ligand-bound complexes suggest a slightly increased positive Mulliken charge of +0.54 centered on the amidino group of benzamidine relative to that of *N*-amidinopiperidine (+0.50), which might contribute to the increased potency of the former inhibitor, although the aliphatic cyclic hydrocarbon should be less costly to desolvate than the aromatic moiety.

**Planar and pyramidal geometry at the guanidino nitrogen.** The ultra-high 0.86 Å resolution of the *N*-amidinopiperidine-trypsin

X-ray structure determined at 100 K allowed us to uncover an intriguing phenomenon. During refinement, it became evident that *N*-amidinopiperidine adopts two alternative conformations (Fig. 3a, b). These do not only differ in the puckering of the piperidine ring but also in the local geometry surrounding the endocyclic guanidino nitrogen. According to a least-squares refinement performed with SHELXL[24], the piperidine nitrogen and the three adjacent carbons are almost in one plane in the higher populated conformer (gray molecule in Fig. 3c, 63% occupancy). Unexpectedly, the alternative conformer is not planar at this endocyclic nitrogen but rather pyramidal with an improper torsion angle of 20.7° ± 4.0° spanned by these atoms (yellow molecule in Fig. 3c, 37% occupancy; this improper torsion corresponds to an out-of-plane deviation of the endocyclic nitrogen of 0.29 ± 0.06 Å). Our trypsin structure complexed with a larger D-Phe-Pro analog of *N*-amidinopiperidine confirmed this pyramidalization as also there the head group occurs exclusively in the non-planar state (Fig. 3d, e). Remarkably, in the case of thrombin binding, *N*-amidinopiperidine did not show any evidence of pyramidalization (PDB code 4UE7)[25].

In order to understand why the pyramidal state can energetically compete with the planar geometry while bound to the S$_1$ pocket of trypsin, despite a likely rupture of conjugation between the orbitals at the piperidine nitrogen and the amidino π-system, we performed a Cambridge Structural Database (CSD) search and QM calculations. The three-fold non-H substituted guanidino N-atoms in small-molecule crystal structures are not always strictly in plane with the three surrounding atoms (improper torsion of −0.6 ± 7.4° (Fig. 3f, some examples are shown in Fig. 3g–i) compared to the much lower scatter of

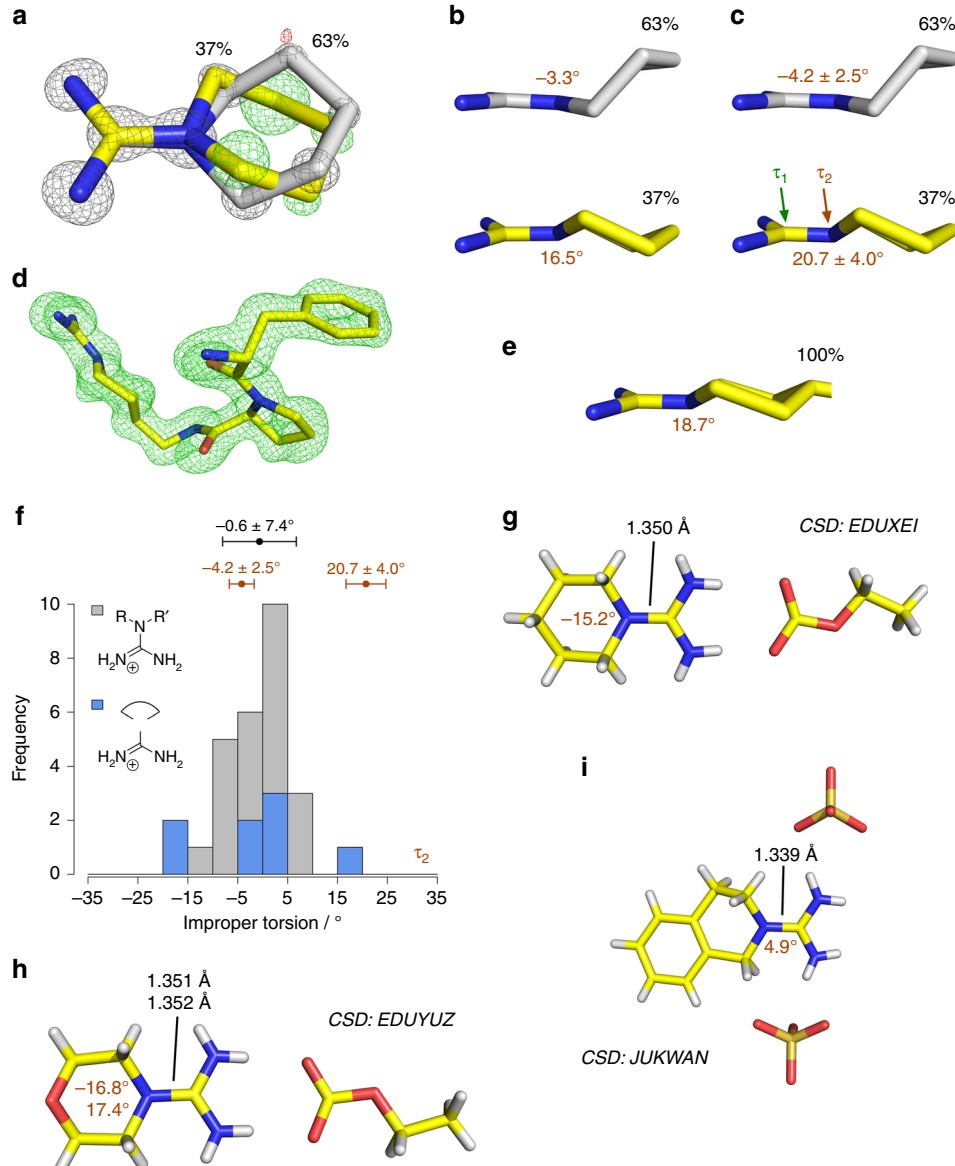

**Fig. 3** Alternative *N*-amidinopiperidine states. All ligands are labeled with their refined occupancies (black percentage values) and with improper torsion angles defining the planarity of the piperidine nitrogen and its environment (in brown). **a** Observation of two conformations for *N*-amidinopiperidine. After *N*-amidinopiperidine had been modeled in the 100 K X-ray structure, an alternative state of this ligand could be identified in the mF$_o$-DF$_c$ map. While this map is depicted at the 3σ level in green, the 2mF$_o$-DF$_c$ map at the refinement state prior to the inclusion of the second conformer is shown in gray at 3σ. **b** Differences in the piperidine nitrogen state according to the Phenix refinement. **c** Differences in the piperidine nitrogen state according to a SHELXL least-squares refinement. The estimated standard deviation is indicated after the ±sign. **d** Difference electron density at 3σ prior to the inclusion of a *D*-Phe-Pro-based analog of *N*-amidinopiperidine into the corresponding X-ray structure. **e** Head group of the ligand shown in panel D displaying a non-planar piperidine nitrogen surrounding. **f** Distribution of improper torsion angles centered on the guanidino nitrogen atoms carrying three non-H substituents (τ$_2$ in **c**) derived from the above-mentioned CSD search. The gray and blue bars represent compounds with two substituents that are separated or linked by a cyclic bridge, respectively. Means and standard deviations for all compounds resulting from the CSD search are given above the bars in black and are compared to the corresponding values for both conformers from the SHELXL refinement of the X-ray trypsin:*N*-amidinopiperidine structure (brown). H-substituted guanidines have been omitted from the CSD search intentionally because X-ray crystallographic coordinates of H-atoms are error-prone. The CSD search, thus, resulted in the limited number of N = 28 hits. The distribution of τ$_1$ values is shown in Supplementary Figure 1. **g-i** X-ray structures of 6-membered guanidino-group containing rings. Together with 5-membered analogs (Supplementary Figure 2), these structures represent the subset of the CSD search, which is highlighted in blue in **f** and Supplementary Figure 1. For the structure with the CSD-code EDUYUZ two values are shown for the improper torsion and bond length since two molecules are present per asymmetric unit of the crystal structure[63]

0.2 ± 1.0° for the improper torsion centered on the associated guanidino carbon atoms (Supplementary Figure 1)). Remarkably, pyramidalization at N occurs in cases where the guanidino group is involved in strongly polarizing bidentate salt bridges to a carboxylate ion in the crystal structures (cf. Figure 3g, h). Examples lacking this contacting salt bridge remain much closer

to a planar arrangement (Fig. 3i and Supplementary Figure 2). The former examples comprising a strongly polarizing salt bridge resemble very much the situation found in the trypsin S$_1$ pocket between *N*-amidinopiperidine and Asp189 (Fig. 2b). They help to explain the observed pyramidalization of the first conformer. Indeed, an unrestrained in vacuo DFT geometry optimization

starting from the pyramidal geometry of the isolated *N*-amidinopiperidine molecule (improper torsion of 15.7°) resulted in a planarization of the molecule (improper torsion of 1.2°), whereas an *N*-amidinopiperidine:aspartate contact pair remained in a pyramidal state (improper torsion of 17.8°). Similarly, the pyramidal state is also maintained after a short QM/MM simulation of the trypsin-bound state while it relaxes to the planar state in the absence of trypsin (s. Supplementary Note 2 and Supplementary Figure 3). The above QM geometry optimizations also suggested that the orbital at the piperidine nitrogen hosting the free electron pair has more of a p-character in the planar than in the pyramidal state (Supplementary Figure 4). Thus, with planar geometry, the nitrogen can better interact with the π-system of the adjacent amidino group, whereas the pyramidal geometry is associated with a decoupling of the amidino group from the neighboring piperidine system. In consequence, the overall Mulliken charge of free *N*-amidinopiperidine is reduced from +1.00 to +0.64 in the *N*-amidinopiperidine:aspartate model complex. At this point, it remains speculative whether Asp189 in thrombin exhibits slightly different charge properties that do not allow population of the pyramidal conformer of *N*-amidinopiperidine moiety as observed in trypsin (see below).

**Changes in the hydrogen-bonding network upon ligand binding.** Of special interest are changes in the hydrogen-bonding network accompanying ligand association upon complex formation. Water W2 contacting either W1 or W3 in the uncomplexed state (Fig. 1a), is completely displaced by both investigated ligands (Fig. 2). Water W3 gets also, at least largely, displaced upon ligand binding. Residual density, however, indicates that the former W3 site is still partly occupied albeit we refrained from refining W3 in the structural models due to low occupancy or high *B*-value when assuming full occupancy (Fig. 2a, c). Interestingly, the presence of W1 is compatible with ligand binding although the immediate environment of this water molecule is significantly altered (Fig. 4). This change stimulates W1 to adopt only one of the two distinguished configurations observed in the uncomplexed state (Fig. 4a). In the ligand-bound state, this water accepts a hydrogen bond from the amidino group of each ligand (Fig. 4b, c). Apart from this interaction, W1 has a lack of additional well-located interaction partners. A second contact is established to Val227(C=O). The water hydrogen cannot approach the CO free-electron pair frontally suggesting that this H-bond remains geometrically suboptimal. The second water hydrogen points toward the accessible π-face of Tyr228, which is likely not an ideal acceptor. In contrast to the situation in the uncomplexed state, where this water is characterized by enhanced residual mobility, it clearly looses rotational degrees of freedom upon ligand accommodation. This will likely result in an entropic penalty that seems not to be compensated by strong enthalpic interactions. Moreover, as MD simulations suggest, W1 can easily exchange with other water molecules located in the S₁ pocket in the uncomplexed state resulting in additional translational mobility, whereas the exchange is strongly hindered once benzamidine or *N*-amidinopiperidine are present (Supplementary Figure 5). In the uncomplexed state, W1 generally escapes the primary hydration layer of Tyr228 after 6 ps. In addition, the rotational states decay in the range of 3 ps once the W1 hydration site is populated, which confirms the crystallographic observation of multiple rotational states for this water molecule. Consequently, we consider W1 to be trapped upon ligand binding with suboptimal energetic properties. In contrast, chlorobenzyl or chlorothiophenyl-type ligands completely displace W1 from the S₁ pocket of trypsin-like serine proteases such as thrombin and

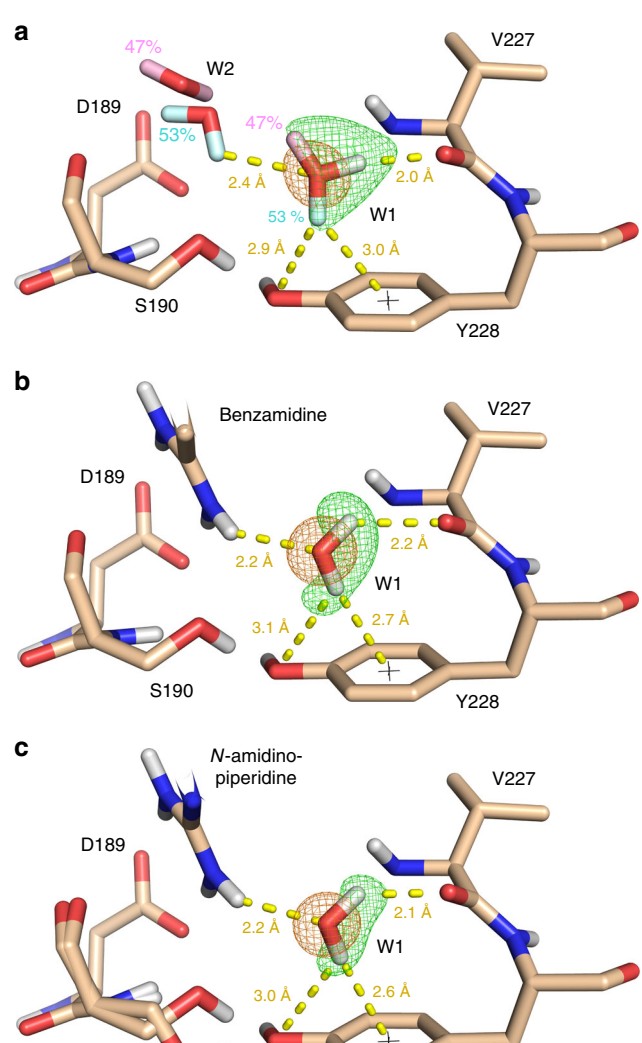

**Fig. 4** Orientation of a water molecule trapped above Tyr228 upon inhibitor binding. X-ray $2mF_o$-$DF_c$ maps for water W1 are depicted at 1.5 σ in orange while the green mesh represents the neutron $mF_o$-$DF_c$ omit map calculated after removal of the W1 deuterium atoms at sigma levels of 3.5 for the uncomplexed and 5.0 for the ligand-bound structures. Selected amino acids are shown in light brown stick representation. **a** XN structure of trypsin in its uncomplexed form. **b** XN structure of the trypsin:benzamidine complex. **c** XN structure of *N*-amidinopiperidine-bound trypsin

FXa. Frequently, such ligands are strong enthalpic binders and have a potency that is comparable to compounds containing basic head groups although they cannot form the favorable salt bridge with Asp189[6]. In addition to reduced ligand desolvation costs, we believe that the unexpected high affinity of chloro-containing ligands can be explained by the capture of W1 in an energetically suboptimal state in the case of basic ligands.

**Reaction pathway of ligand binding and solvent rearrangement.** Our combined XN refinements uncovered structural changes upon benzamidine or *N*-amidinopiperidine binding to trypsin's S₁ pocket, and in particular, rearrangements of the H-bonding network along with modulations of the rotational states of the active-site waters. Crystallography, however, does not suggest straightforward the reaction pathways along which these changes occur. We applied metadynamics, as enhanced MD sampling technique[26], to simulate the benzamidine and

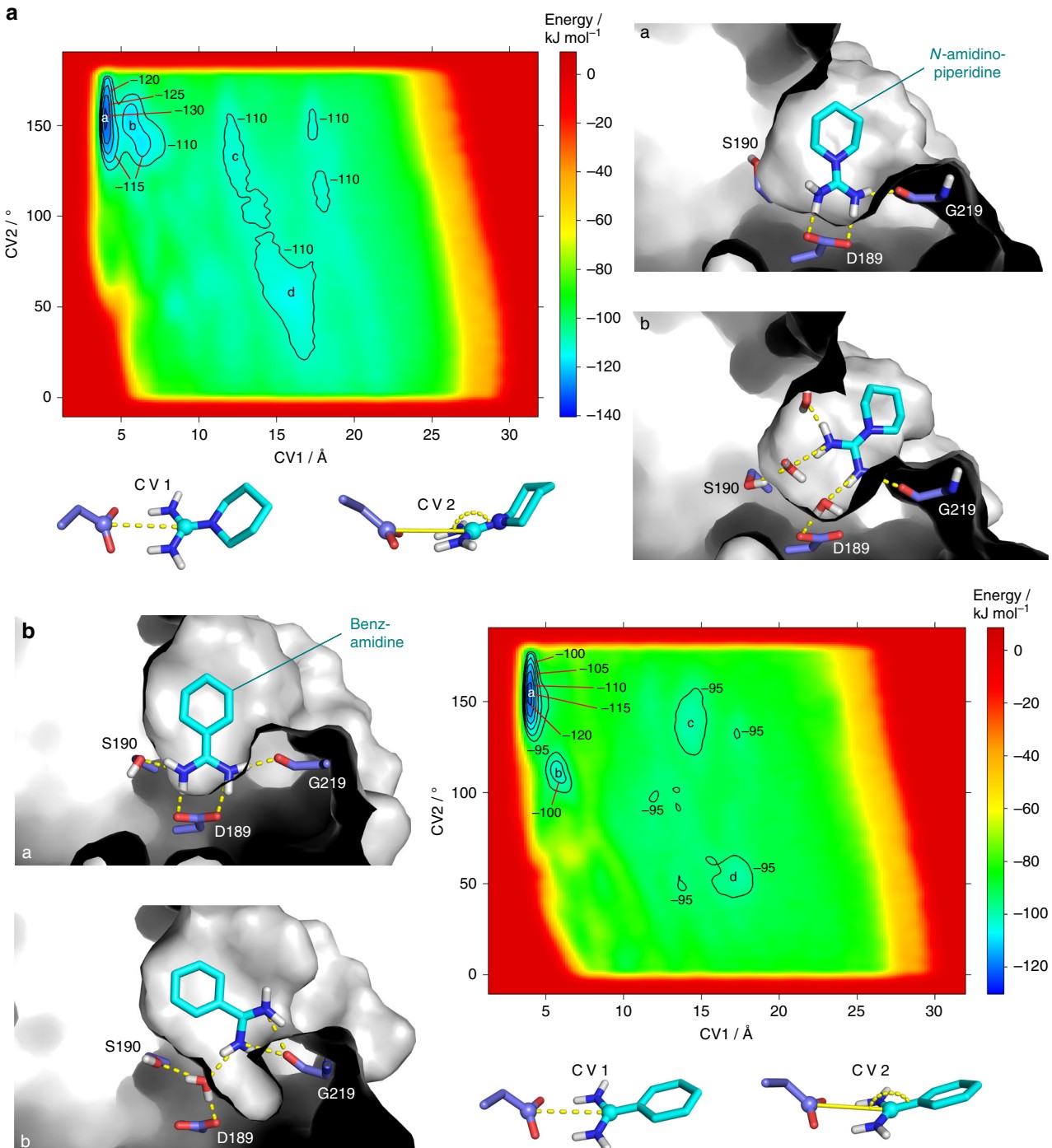

**Fig. 5** Free energy surfaces of trypsin:inhibitor complexes projected onto CV1 and 2. Explicit contour levels in kJ mol$^{-1}$ are highlighted in the trypsin:*N*-amidinopiperidine (**a**) and trypsin:benzamidine (**b**) energy diagrams by black lines while gradual energy changes are reflected by color. The respective collective variables are defined below each free energy profile. Representative structural snapshots corresponding to the major energy minima specified with the letters a and b in the FES are presented next to the energy diagrams with the ligands shown in cyan, while structures for the additional less prominent energy minima are shown in Supplementary Figure 8

*N*-amidinopiperidine binding and unbinding event and to reconstruct the associated free-energy surfaces (FES) relative to two collective variables (CV1, CV2, see also Fig. 5). CV1 defines the distance between the carboxylate carbon of Asp189 and each ligand's amidino carbon. CV2 describes the angle between these two atoms and the adjacent *ipso*-ring atom attached to the amidino group (Fig. 5). For both ligands, we observed several binding and unbinding events (Supplementary Figure 6 and 7), which allowed the reconstruction of the FES in both instances (Fig. 5).

The energetic landscapes are very similar for *N*-amidinopiperidine and benzamidine. The deepest minimum a corresponds to the final protein-ligand complex with formed salt bridge to Asp189 (Fig. 5). In minimum b, the ligands are rotated slightly out of the S$_1$ pocket and remain H-bonded to Gly219(C=O) and water molecules intercalating between inhibitor and Asp189. Interestingly, the orientations of *N*-amidinopiperidine and benzamidine differ in this intermediate state (Fig. 5). Taking in both cases the energy level of minimum b as a reference, benzamidine

is by 6.0 kJ mol$^{-1}$ more stable than $N$-amidinopiperidine in the respective final minimum a. This nicely corresponds to the relative $\Delta G°$ differences of 4.7–6.8 kJ mol$^{-1}$ measured by our ITC protocols (Supplementary Table 2). The consistency with our ITC data is also visible in Supplementary Figure 6E and 7D. A closer inspection of the trajectories revealed that the inhibitors visit frequently the S$_1$ pocket but also stay at the more remote S$_2'$ and S$_{3/4}$ pockets (Supplementary Figure 6B and 7A). These intermediate arrangements correspond to shallow energy minima. $N$-amidinopiperidine binds in two different orientations to the S$_{3/4}$ pocket (minima c and d, Supplementary Figure 8A). The same holds for benzamidine except that this molecule seems to be more easily accommodated in the S$_2'$ pocket compared to $N$-amidinopiperidine (Supplementary Figure 8B).

Since minima a and b are structurally related and directly adjacent on the FES, it appears likely that b represents an intermediate visited at an early stage during dissociation or shortly before reaching the final state during ligand association, respectively. For benzamidine, a similar proposal was made by Gervasio et al.[27]. The barrier between minima a and b could, therefore, represent a kinetic bottleneck along the ligand (un) binding path. The dissociation barrier is much higher for benzamidine (30.7 kJ mol$^{-1}$) than for $N$-amidinopiperidine (16.8 kJ mol$^{-1}$). Similarly, the association barrier, estimated from the FES, is 11.6 kJ mol$^{-1}$ for benzamidine, while only 3.7 kJ mol$^{-1}$ have to be overcome for $N$-amidinopiperidine. These data suggest that the lower affinity of $N$-amidinopiperidine mainly suffers from an enhanced off-rate compared to the more potent benzamidine (experimental $k_{\text{off}} = 6 \times 10^2$ s$^{-1}$ according to ref. [28]; Supplementary Figure 6E and 7D). From a geometrical point of view, the energy minima a and b differ by the intercalation of water molecules between the ligand's amidino group and Asp189. Thereby the central bidentate salt bridge is destroyed, which helps reducing the dissociation energy barrier. This is definitely an essential step toward inhibitor dissociation and may account for the structure–kinetic relationship between the two studied ligands. In return, the removal of the final solvation layer between the polar amidino group and Asp189 likely imposes an important kinetic barrier to the association step. Interestingly, our previously reported XN structure of 2-aminopyridine bound to trypsin resembles the intermediate state b identified for benzamidine with one water molecule located between the ligand and Asp189[14]. Moreover, simulations of the binding kinetics of other proteins suggest that dehydration of the binding site and in particular, the removal of the final solvation layer has a major impact on the ligand association barrier[29–32]. For instance, a single water molecule separates biotin and Asp128 of streptavidin[29], similar to our situation in trypsin. While inspecting individual unbinding events along the metadynamics trajectories (s. Supplementary Movies 1 and 3), we found that the water molecule, incipiently intercalating between inhibitor and Asp189, always comes from the water reservoir beyond Asp189. It usually emanates from a location close to a hydration site where W3 is detected (Fig. 1 and Supplementary Figure 9). Although not sufficient for explicit placement in the final refinement model, residual electron and nuclear density are visible for a water molecule at that site in both ligand-bound XN structures (Fig. 2a, c). Consistently, the plain MD simulation of the trypsin:$N$-amidinopiperidine complex indicates the intermittent presence of a water molecule at this hydration site (Supplementary Figure 5A) and renders a key role to this water molecule during dissociation or, in reverse, association. In subsequent steps of dissociation, a second water molecule further promotes the separation of ligand and Asp189. This water molecule also comes from internal cavities, most of the time from the above-mentioned water reservoir. Subsequently, additional water

molecules penetrate into the binding site most frequently via the main entrance of the S$_1$ pocket, ultimately leading to the fully dissociated state and a hydrated ligand-free S$_1$ pocket (Supplementary Movie 1 and Supplementary Movie 3). Since the main portal to the S$_1$ pocket is occluded by the ligand in its fully bound state, it is obvious that the water molecules from the reservoir behind Asp189 are very important to initiate the dissociation process. If these water molecules would be completely absent, very slow off-rates would likely result. This is either linked to an energetically highly unfavorable salt-bridge rupture without formation of compensating H-bonds with intercalating waters, or to a process in which water molecules would have to squeeze into the S$_1$ pocket via the main entrance, which is occluded by the ligand itself. Such an unlikely situation would drastically reduce the enzymatic activity of trypsin leading to very slow product dissociation. We therefore believe that the internal water reservoir is crucial and has evolved to facilitate the latter step. Consistently, a theoretical study by Schmidtke et al. concluded that water-shielded H-bonds result in a kinetic trap[31]. Importantly, all steps described above for the unbinding event apply in reversed sequence to the association process (Supplementary Movie 2 for $N$-amidinopiperidine).

## Discussion

It is extremely difficult to collect precise experimental information about hydrogen atoms that discloses the establishment of H-bonding networks and the orientational as well as rotational properties of water molecules in protein binding pockets before and after ligand binding along with mutual transformations upon complex formation. Nevertheless, particularly the impact of water on the thermodynamic inventory of ligand binding is decisive. Without this information, our view on the ligand-recognition process must remain incomplete. Although water has been discussed to contribute to protein-ligand interactions in many ways, explicit experimental evidence supporting these concepts is rare. In the present joint experimental and computational study, we determined neutron and X-ray structures of trypsin in its uncomplexed, benzamidine and $N$-amidinopiperidine-bound state. Data collected at ultra-high resolutions enable a very accurate description of the structural states prior and subsequent to the ligand-binding event disclosing a high percentage of H-atom positions and water orientations (for statistics see Supplementary Tables 4 to 7). These experimental data were supplemented by metadynamics simulations describing the ligand association and dissociation process.

ITC data and the simulations suggest that benzamidine binds more tightly to trypsin compared to $N$-amidinopiperidine, likely based on a slower off-rate. We propose that the formation of the trypsin:$N$-amidinopiperidine complex, relative to the one with benzamidine, is enthalpically less favorable due to a decreased electrostatic interaction energy between ligand and Asp189. The positive charge on the amidino group is significantly reduced for $N$-amidinopiperidine compared to benzamidine according to our QM/MM simulations. We believe that this is caused by the additional delocalization of the charge onto the piperidine nitrogen, which is located farther from Asp189. In case of thrombin, $N$-amidinopiperidine is more potent than benzamidine, thereby creating a selectivity advantage for this ligand. Strikingly, thrombin, in contrast to trypsin, harbors additional negatively charged residues (e.g. Glu192) close to the entry of the S$_1$ pocket, which are absent in trypsin (Gln192), and thus might explain why the shift of the positive charge away from the position of Asp189 may be tolerated for thrombin. The reduced electrostatic interactions toward the upper rim of the S$_1$ pocket in trypsin (Gln/Glu exchange between trypsin/thrombin) may even

cause the very surprising structural phenomenon observed in our study: In contrast to thrombin, *N*-amidinopiperidine does not only bind to trypsin with planar geometry, but also populates a pyramidal hybridization state at the piperidine nitrogen. This pyramidalization goes along with a decoupling of the piperidinyl moiety from the amidino group, thus losing electronic conjugation. Simultaneously, the positive charge likely accumulates more strongly on the amidino group and therefore enables an energetically more favorable interaction with Asp189. This compensates for the loss of π-orbital overlap and explains the existence of two configurational isomers at the endocyclic piperidine nitrogen. Furthermore, thrombin hosts a regulatory sodium ion in 5.7 Å distance from Asp189. Most likely, this positive charge in close proximity influences the charge distribution on the crucial aspartate residue in thrombin. It can be speculated that this charge attenuation further explains why in case of thrombin the pyramidalization at the endocyclic guanidino nitrogen is not observed (PDB code 4UE7)[25].

It is insufficient to only consider the final ligand-bound state to fully understand how a ligand gains affinity toward a certain target. The hydration of a ligand and its receptor also plays a fundamental role in the ligand-recognition process. Here, we provide experimental evidence that the water molecules filling the $S_1$ pocket of uncomplexed trypsin are (i) not in ideal H-bonding geometry, e.g. to the carboxylate of Asp189, (ii) not fully saturated by an embedment into an optimal H-bonding network but instead (iii) remain rather mobile and therefore adopt multiple orientations and, in part, locations. As a result, it appears favorable for the penetrating ligand to displace these water molecules from the binding site. Likely, this is essential for efficient substrate and inhibitor binding to trypsin because they suffer already from large energetic costs to be spent for the desolvation of their in most cases charged $P_1$ anchor group in the $S_1$ binding pocket. Apparently, the $S_1$ pocket of the enzyme evolved in a way that Asp189 is not optimally hydrated in order to enable efficient substrate binding. In his review, Steve Homans suggested that suboptimally hydrated binding pockets might in fact be a more general feature in proteins[2]. Our experimental data collected for trypsin clearly supports this hypothesis and we hope to trigger similar studies with other proteins to further show the generality of this concept.

Not all water molecules are displaced from the $S_1$ pocket when the investigated amidino ligands bind to trypsin. We experimentally show that the water molecule located above Tyr228 (W1) is trapped upon complex formation and frozen in one particular configuration while it is scattered across at least two orientations in the uncomplexed state. We suggest that this phenomenon represents an entropic disadvantage for ligands that are not able to displace this water molecule. In agreement with computational predictions[6], this also explains why uncharged inhibitors, e.g. of the m-chlorobenzyl or chlorothiophenyl-type are able to compete in potency with the positively charged head groups such as benzamidine or *N*-amidinopiperidine. They achieve complete dehydration of all water molecules from the $S_1$ pocket and obtain a more enthalpy-favored binding signature as they displace the additional, rather mobile W1 water molecule located above Tyr228. Overall, this provides an enthalpic advantage as the release of this mobile, entropically favored water molecule to the bulk does not enhance entropy but enthalpy due to the formation of better H-bonds in the surrounding water phase. Furthermore, chloro-substituted ligands establish van der Waals and halogen bonding interactions that are likely enthalpically favorable[33].

Ultimately, water molecules do not only influence binding by flooding the uncomplexed state of the binding pocket and mediating protein-ligand interactions, but they also seem to have a critical impact on the ligand association and dissociation process itself. Our metadynamics simulations suggest an intermediate state on the free-energy surface along the association and dissociation pathway that differs from the global energy minimum by the intercalation of a water molecule between the amidino group of the ligand and Asp189. A detailed analysis of our metadynamics trajectories prompted us to suggest that water molecules involved in this process become stored in or originate from a protein-internal reservoir cavity, which likely evolved to facilitate the accommodation of the enzymatic substrate or release of the generated product.

The protein-ligand binding event is influenced by both, the states before and after binding. Often we merely concentrate on the states following binding, usually as these are easier or sometimes even solely accessible by experimental approaches. Consequently, the situation prior to binding is only vaguely considered or remains completely unknown. This especially concerns the detailed water structure prior to ligand binding and its changes upon protein–ligand complex formation. In the case of X-ray diffraction, the proper water pattern is usually impossible to resolve sufficiently and no insights from an experimental point of view can be given on the rotational and translational states of water molecules in the different states. In the presented series of highly resolved neutron and X-ray structures, we elucidate the imperfect hydration of Asp189 in the uncomplexed protein, a residue directly involved in substrate recognition in the S1 pocket of trypsin-like serine proteases. Supposedly, this solvation pattern is a prerequisite for potent substrate and, similarly, inhibitor binding to these proteases. Furthermore, changes in the ordering states of the water molecules, which mediate contacts between protein and ligands, are observed and correlate with thermodynamic binding data. A surprising pyramidalization of the endocyclic guanidino nitrogen of *N*-amidinopiperidine experienced upon binding to trypsin, but not to thrombin, indicates differences in the charge concentration on the Asp189 residue between both proteases, which might influence substrate recognition and selectivity.

Although these conclusions are drawn from this special case of trypsin, we believe that they are valid for a much broader range of protein-ligand complexes. We hope that our analysis triggers similar in-depth studies that collect experimental evidence to stimulate the development of computational models. Thereby, we will further complement our general view of protein-ligand recognition in order to facilitate a more reliable target-oriented design of novel drugs required to face unmet medical needs.

## Methods

**Neutron and X-ray crystallography**. In the following, we will describe the crystallization, refinement and modeling strategy for the neutron and XN structures of uncomplexed trypsin and trypsin in complex with benzamidine or *N*-amidinopiperidine, respectively, together with an additional X-ray structure of a complex formed with the *D*-Phe-Pro analog of *N*-amidinopiperidine. Furthermore, Supplementary Figure 10 highlights how the important catalytic triad has been modeled. Importantly, His57 was identified to exist in two alternative states, in which $N_\delta$, pointing toward the deprotonated Asp102, is always protonated while $N_\epsilon$ carries a deuteron with only partial occupancy (Supplementary Figure 10). This confirms the general ability of His57 to act as a base during catalysis[34].

**Crystallization and H/D-exchange**. All crystallographic experiments and structure determinations were performed following an approach that is consistent with our earlier work[14]. Crystals were prepared from bovine β-trypsin purchased from Sigma Aldrich (product numbers T8003 and T1426). For ligand-bound crystals, trypsin was dissolved in a buffer composed of 15 mM HEPES pH 7.0, 7.5 mM $CaCl_2$ and 25 mM benzamidine or *N*-amidinopiperidine to yield a final protein concentration of 60 mg ml$^{-1}$ as evaluated by the weight-in mass of lyophilisate. In the case of crystal growth for the neutron diffraction studies 75 instead of 25 mM benzamidine were used. For preparation of apo-crystals, the buffer contained no ligand but a higher $CaCl_2$ concentration of 50 mM. Since crystals for neutron diffraction studies need to have a volume in the mm$^3$ range, special equipment was

used to grow such large crystals. In particular, we applied the sitting-drop vapor diffusion method at a larger than usual scale with drops consisting of 100 µl trypsin or trypsin-ligand solution and 100 µl mother liquor, which was composed of 0.1 M HEPES pH 7.5, 0.2 M $(NH_4)_2SO_4$ and 13.0% (w/v) PEG 8000 for trypsin:benzamidine or 16.0% (w/v) PEG 8000 in the case of trypsin:$N$-amidinopiperidine and the uncomplexed form of trypsin. The crystallization setups were prepared either using the Sandwich-box system from Hampton Research (product number HR3-136; used in the case of trypsin:$N$-amidinopiperidine) or center-well dishes kindly provided by Corning (product number 353037; used in the case of trypsin:benzamidine) or purchased from VWR (product number NUNC150260; used in the case of apo-trypsin). Crystals were then grown at 4 °C. In order to reduce adherence of the large trypsin:$N$-amidinopiperidine crystals to the Sandwich-box wells, wells were pre-filled with the high-density Fluorinert® FC-70 oil purchased from Sigma Aldrich (product number F9880). Applying these procedures, large crystals for neutron diffraction data collection could be obtained. Crystal volumes were 1.9, 1.8, and 6.5 mm³ for uncomplexed trypsin, trypsin:$N$-amidinopiperidine and trypsin:benzamidine, respectively. Following their growth to full size, these crystals were subjected to gradual H/D-exchange at 17 °C. The $D_2O$-based solution used for the stepwise exchange consisted of 0.1 M HEPES pD 7.9, 0.2 M $(ND_4)_2SO_4$, 21.5–23.0 % (w/v) PEG 8000, 0.1 mM $CaCl_2$ and, in the case of ligand-bound crystals, 25 mM $N$-amidinopiperidine or benzamidine. Since the $pK_{W,25\ °C}$ of $D_2O$ is 14.9 vs. 14.0 for $H_2O$, the pD of 7.9 of this solution corresponds to a pH of ~7.5[35]. The crystal of uncomplexed trypsin stayed 47 days in the fully deuterated solution before it was mounted in a hollow round quartz capillary purchased from CM Scientific. Data collection begun 54 days after crystal transfer into the capillary resulting in additional time for H/D-exchange via vapor diffusion in which a small drop of deuterated mother liquor added to the capillary before sealing acted as deuteration source. Similarly, the trypsin:$N$-amidinopiperidine crystal was soaked in deuterated liquid for 21 days and mounted in a quartz capillary 15 days prior to the start of data collection. Due to its even larger size, the trypsin:benzamidine crystal was mounted in a quartz NMR tube 3 days before initiation of data collection. This was done following a 14 day-long incubation period in deuterated mother liquor.

For X-ray data collection, smaller crystals obtained from crystallization trials prepared as mentioned above or from setups in standard 24-well sitting-drop crystallization plates were used in order to avoid X-ray absorption effects expected for large crystals[11]. The mother liquor was composed of 0.1 M HEPES pH 7.5, 0.2 M $(NH_4)_2SO_4$ and 16–20 % (w/v) PEG 8000. Prior to data collection at room temperature, each crystal was subjected to H/D-exchange using the above mentioned procedure and then mounted either in a quartz capillary (uncomplexed trypsin, trypsin:$N$-amidinopiperidine) or in a MicroRT™ capillary (trypsin: benzamidine) purchased from MiTeGen (product number RTSK-1). For data collection under cryogenic conditions, one H/D-exchanged crystal of each system was mounted in a nylon loop and briefly dipped into a cryoprotectant solution containing 70 mM HEPES pD 7.9, 130 mM $(ND_4)_2SO_4$, 19 % (w/v) PEG 8000, 25 % (w/v) PEG 400, 70 µM $CaCl_2$ and, if required, 17 mM of the corresponding ligand followed by immediate flash-cooling in liquid nitrogen. For determination of non-deuterated apo-trypsin crystal structures at 100 and 295 K, three uncomplexed trypsin crystals were prepared using each of the two protocols described above except that no H/D-exchange was performed.

In order to determine a structure of trypsin in complex with a $D$-Phe-Pro analog of $N$-amidinopiperidine (compound **1** in Supplementary Table 1), bovine trypsin obtained from Sigma Aldrich (product number T8003) was dissolved at a concentration of 45 mg ml⁻¹ in 15 mM HEPES pH 7.0, 7.5 mM $CaCl_2$, 3 mM ligand and 6% (v/v) DMSO. Applying the hanging-drop vapor diffusion method with a 2 µl drop size, crystals grew in a solution containing 0.1 M HEPES pH 7.5 and 2.0 M $(NH_4)_2SO_4$. A diffraction-quality crystal was transferred into a solution consisting of 0.1 M HEPES pH 7.5, 2.0 M $(NH_4)_2SO_4$, 3 mM ligand, 6% (v/v) DMSO and 27% (v/v) glycerol and subsequently flash-frozen in liquid nitrogen.

**Neutron diffraction data collection.** Crystallographic experiments using neutrons were performed at the BIODIFF instrument operated by the FRM II and JCNS at the Heinz Maier-Leibnitz Zentrum (MLZ, Garching, Germany). Diffraction data were collected at 295 K using a monochromatic neutron beam with the wavelength adjusted to a value between 2.67 and 2.68 Å. Exposure times were set to 50 min and the oscillation range to 0.4°. After a first series of images had been collected, the investigated crystal was tilted relative to the rotation axis prior to a second pass of data collection significantly enhancing completeness of the data. To further increase completeness, 23 additional images were collected from the uncomplexed trypsin crystal during a third run using the same measurement settings. For the trypsin:benzamidine system, a short third pass was conducted with an exposure time of only 3 min and an oscillation range of 0.7° to collect also the strongest low resolution reflections. Data were indexed, integrated and scaled using HKL2000 with an empirical absorption correction applied for both ligand-bound structures[36]. Five percent of all reflections were assigned to the $R_{free}$ subset of data. The resulting data collection statistics are shown in Supplementary Table 4.

**X-ray diffraction data collection.** X-ray crystallographic experiments at 295 and 100 K were performed at the EMBL beamline P14 at the DESY (Hamburg, Germany). Collection of room temperature data of very high resolution was enabled by

the fast PILATUS 6 M detector in combination with the helical scanning technique, which was applied in order to keep radiation damage minimal. Data were processed using XDS[5]. Data collection statistics are given in Supplementary Tables 5 (100 K) and 6 (295 K). For a closer investigation of the water structure in the $S_1$ pocket of uncomplexed trypsin, we also collected three additional X-ray diffraction data sets from non-deuterated uncomplexed trypsin crystals each at 295 K and at 100 K. Data were processed and structures refined using our PHENIX-based[37] automated pipeline tool[20]. The crystal of trypsin in complex with the $D$-Phe-Pro analog of $N$-amidinopiperidine underwent X-ray exposure on beamline 14.1 at BESSY (Berlin, Germany) with subsequent XDS-based data processing. $R_{free}$ flags were randomly assigned and account for 5% of all data. The respective statistics are listed in Supplementary Table 5.

**Refinement against neutron diffraction data.** Phases were obtained via molecular replacement using the PDB entry 4I8H as search model. Refinement has been performed against structure factor amplitudes using PHENIX and five macrocycles in each refinement[37]. In the first round of refinement, simulated annealing has been used to remove potential model bias originating from the search model. H and D atoms were only included in the model when clearly visible at the ±2.7 σ level in the $mF_o$-$DF_c$ map. Please note that these two hydrogen isotopes can be clearly differentiated by neutron crystallography since they are characterized by neutron scattering lengths with opposite sign, meaning that H atoms cause negative peaks in the $mF_o$-$DF_c$ map whereas D atoms yield positive peaks. At polar and thus exchangeable sites, the occupancies of simultaneously present H and D atoms were refined to sum up to 1 while the coordinates and atomic displacement parameters (ADP) of these atoms were constrained to the same values. Refinement statistics are given in Supplementary Table 4.

**Refinement against X-ray diffraction data.** The initial model has been obtained by the molecular replacement technique using the PDB entry 4I8H as search model. Refinements of the models deposited in the PDB have been performed against structure factor amplitudes using PHENIX and five macrocycles in each refinement round. In order to remove potential model bias originating from the search model, simulated annealing has been performed in the first round of refinement. All hydrogen atoms were modeled as H isotopes since they cannot be distinguished from D atoms by X-ray crystallography. They were only added to the respective model if clearly visible at +2.5σ in the $mF_o$-$DF_c$ map and refined according to the riding model. As an exception, all hydrogens were added to the structural model of trypsin in complex with the $N$-amidinopiperidine analog to improve model quality at the slightly lower resolution. Refinement statistics can be found in Supplementary Tables 5 (100 K) and 6 (295 K).

In the case of trypsin:$N$-amidinopiperidine, also PHENIX[37] was used for refinement to generate the structural model for the data collected at 100 K (PDB code: 5MNN). In order to estimate standard deviations for geometric parameters and in particular the planarity of the ligand's piperidine nitrogen also a SHELXL[24] refinement against intensities was performed. In a first step, the fully refined structure as deposited in the PDB (5MNN) was changed into sequential residue numbering using the PHENIX PDB file editor tool[37]. This file was converted with SHELXPRO, option I into the file format required for SHELXL refinement[38]. As hydrogen atoms can be refined in numerous ways in SHELXL they were not converted from the PDB file, but added later for the protein with the recommended default settings (HFIX). All default commands set by SHELXPRO were maintained with the following changes/additions: The number of conjugate-gradient refinement cycles was set to 30 and included the $R_{free}$ flag (CGLS 30 -1). The resolution was set to include all data (SHEL 999 0.1) and enhanced rigid bond restraints were applied (RIGU)[39]. While additional restraints for the sulfate molecules were added by using SHELXPRO, option J, the ligand $N$-amidinopiperidine was refined freely except for two restraints. One restraint was set in order to ensure equal lengths for symmetrically equivalent bonds within the piperidine ring in ligand conformations A and B (SADI). The second restraint was designed to guarantee the planarity of the guanidine (plane defined by the central carbon and the three attached nitrogens) (FLAT 0.05), which was justified by the CSD search presented in Supplementary Figure 1. After the first SHELXL refinement, the $R$-value converged to $R_1 = 10.8\%$ for $F_o > 4σ(F_o)$ and 11.8% for all data; $R_{1,free} = 12.4\%$ for $F_o > 4σ(F_o)$ and 13.3% for all data. In the second refinement, again 30 cycles of conjugate gradient refinement were performed, while hydrogen atom refinement was activated for the protein with default settings (HFIX). Now the $R$-values converged to $R_1 = 10.0\%$ for $F_o > 4σ(F_o)$ and 11.0% for all data; $R_{1,free} = 11.5\%$ for $F_o > 4σ(F_o)$ and 12.4% for all data, which is very comparable to the values reported for the PHENIX refined structure. The next SHELXL refinement was performed against all data (including the ones previously used for $R_{free}$ calculation) for 30 cycles. Finally, a single full matrix cycle (L.S. 1) using damping (DAMP 0 0) was performed while keeping the anisotropic displacement parameters fixed (BLOC 1). For this refinement, all restraints for protein and ligands were disabled using the REM setting for all relevant lines.

**Joint refinement against neutron and X-ray diffraction data.** In addition to the above-described individual refinements against neutron or X-ray data, joint X-ray/neutron (XN) refinements were performed to further increase model quality. For

this purpose, diffraction data from the highly isomorphous crystals used for neutron and X-ray crystallographic experiments at ambient temperature were used (Supplementary Tables 4 and 6). The same free reflections as for the neutron data were also selected for the here added X-ray data to ensure that the $R_{free}$ values maintain their significance. Refinements were started from the respective finalized neutron structure. The corresponding nuclear distances were used as restraints for all bonds involving H or D atoms. Supplementary Figure 10 illustrates how the catalytic triad and Ser214 have been modeled. Further information and refinement statistics are depicted in Supplementary Table 7.

**Searches in the CSD**. CSD searches of guanidine structures with the generic formula $RR'N–C^+(NH_2)_2$, where R and R′=C (residue connected via a carbon atom), were performed using ConQuest vs. 1.18 and CSD version 5.37 as of November 2015 with 2 updates[40]. The obtained data were then processed and analyzed using Mercury (vs. 3.8) and plotted using the statistical framework R[41,42]. The following parameters were used in all searches: 3D coordinates determined, only organics, not disordered, no errors and not polymeric. In addition, the crystallographic R value limit has been chosen to be very low (≤0.05) to ensure that all atoms are well resolved resulting in extraction of accurate geometric data.

**Synthesis of N-amidinopiperidine analog**. While benzamidine and N-amidinopiperidine were purchased from Sigma Aldrich (product numbers 434760 and CDS014887, respectively) and the D-Phe-Pro analog of benzamidine (compound **2** in Supplementary Table 1) was available from a previous study[43], the N-amidinopiperidine D-Phe-Pro analog (compound **1** in Supplementary Table 1) was synthesized for the purpose of this study. The $P_1$ building block bis-Boc guanidine was prepared according to the procedure of Das et al.[44]. The subsequent reaction with Boc-D-Phe-Pro-OH[43] to the desired N-amidinopiperidine D-Phe-Pro analog was performed under coupling conditions of amide formation and Boc-deprotection. Boc-D-Phe-Pro-OH (0.25 g, 0.69 mmol), bis-Bocguanine (0.27 g, 0.76 mmol), HOBt hydrate (0.12 g, 0.76 mmol) were dissolved in DMF (5 ml) at 0 °C. Then EDC hydrochloride (0.15 g, 0.76 mmol) and DIPEA (0.24 ml, 1.38 mmol) were slowly added. The mixture was stirred at room temperature for 14 h, diluted with water (5 ml) and extracted with EtOAc (3 × 15 ml). The organic layer was washed with saturated $NaHCO_3$ solution (3 × 15 ml) and saturated NaCl solution (1 × 15 ml), dried over $MgSO_4$, filtered and concentrated in vacuo. Purification by column chromatography (cyclohexan/EtOAc, 3:1 → 1:6) afforded the Boc-D-Phe-Pro derivative (0.29 g, 0.41 mmol, 60%) as a white solid. The resulting compound (0.24 g, 0.34 mmol) was dissolved in TFA (3 ml) and stirred at room temperature for 2 h. The mixture was concentrated in vacuo and the residue was purified by preparative HPLC ($H_2O$/$CH_3CN$/TFA 95:4.9:0.1 → 70:29.9:0.1, 40 min). The hygroscopic D-Phe-Pro derivative (0.22 g, 0.30 mmol, 88%) was obtained as a colorless double TFA-salt with 5.5 $H_2O$. $^1H$-NMR: main conformer (400 MHz, DMSO-$d_6$): δ = 8.33 (s, br, 3H), 7.84 (t, J = 5.9 Hz, 1H), 7.39 (s, 4H), 7.36–7.28 (m, 3H), 7.23–7.18 (m, 2H), 4.37–4.30 (m, 1H), 4.15–4.12 (m, 1H), 3.83–3.79 (m, 2H), 3.52–3.45 (m, 1H), 3.11–3.06 (dd, J = 13.1, 5.8 Hz, 1H), 3.03–2.85 (m, 6H), 2.67–2.60 (m, 1H), 1.76–1.64 (m, 6H), 1.46–1.37 (m, 1H), 1.13–1.01 (m, 2H) ppm. $^{13}C$-NMR: main conformer (100 MHz, DMSO-$d_6$): δ = 170.8, 166.8, 158.9–157.8 (q), 155.6, 134.4, 129.4, 128.5, 127.4, 120.3–111.5 (q), 59.9, 52.0, 46.6, 45.0, 43.3, 36.7, 35.0, 29.4, 28.8, 28.7, 23.6 ppm. MS (ESI+): m/z calcd for $[C_{21}H_{33}N_6O_2]^+$: 401.27; found: 401.36.

**Quantitative $^1H$-NMR experiments**. In order to improve the accuracy of our enzyme kinetic and thermodynamic results, purities of benzamidine and N-amidinopiperidine were analyzed by quantitative $^1H$-NMR (qNMR) spectroscopy. Similar to our recent study[14], NMR data were collected at a field strength of 11.7474 Tesla (500.2 MHz) on a JEOL ECA-500 MHz spectrometer and processed using the Delta NMR Processing and Control Software, version 5.0.0. All spectra were collected in DMSO-$d_6$ and were referenced to the residual solvent peak at δ = 2.50 ppm. Maleic acid (Sigma Aldrich, TraceCERT®, δ 6.26 ppm; product number 92816) was used as certified internal standard. Purities were calculated from the observed integral values of the analyte and reference standard protons considering the concentration of analyte and standard, the number of protons as well as the purity of the standard. These experiments resulted in purities of 97.7 and 96.6% for benzamidine and N-amidinopiperidine, which were subsequently considered during the preparation of ligand solutions for affinity determinations.

**Kinetic enzyme inhibition assay**. Inhibition of bovine trypsin (Worthington Biochemical Corporation, product number LS003707) by benzamidine, N-amidinopiperidine and their D-Phe-Pro analogs was investigated using a fluorescence assay. Following the same approach as used in an earlier study[14], kinetic data were obtained in black 96-well plates using a Fluoroskan Ascent fluorometer (Thermo Fisher Scientific, Vantaa/Finland; $λ_{ex}$ = 355 nm, $λ_{em}$ = 460 nm). The assay was performed at room temperature in a buffer composed of 43 mM Tris pH 7.8, 132 mM NaCl and 0.0086 % (w/v) PEG 8000. The final trypsin concentration was 65 pM. Benzamidine and N-amidinopiperidine were directly dissolved in buffer. The D-Phe-Pro analogs were dissolved in DMSO and further diluted in buffer to yield a final DMSO concentration in the assay well below 1% (v/v). Mes-D-Arg-Gly-Arg-AMC (Mes = mesyl; AMC = 7-amino-5-methylcoumarin) available from

a previous study was used as the substrate[14]. Measurements for the determination of $K_i$ values were performed at 12.5 as well as 25 μM substrate and at four different inhibitor concentrations. The uninhibited trypsin-catalyzed reaction was investigated in parallel at the three substrate concentrations 6.3, 12.5, and 25 μM. The velocities obtained from the slope of progress curves during a time frame of 5 min provided the basis for $K_i$ determination. For this purpose, a non-linear fit to the Michaelis-Menten equation modified for reversible competitive inhibitors was used.

The analogous assay was applied to human thrombin, which was kindly provided by CSL Behring, with Tos-Gly-Pro-Arg-AMC instead of Mes-D-Arg-Gly-Arg-AMC as a substrate (Tos = tosyl) and at an enzyme concentration of 80 instead of 65 pM. The used substrate concentrations were 2.5, 5.0, and 10 μM in case of thrombin compared to 6.3, 12.5, and 25 μM used for trypsin. Triplicate measurements were performed in all cases and $K_i$ values are reported as mean ± standard deviation.

**Isothermal titration calorimetry**. To determine dissociation constants for trypsin:inhibitor complexes, two different isothermal titration calorimetry (ITC) titration protocols was used. The direct titration approach was used for N-amidinopiperidine and benzamidine as well as for the D-Phe-Pro derivative of each of these molecules (ligands **1** and **2** from Supplementary Table 1). Ligand solutions of 2.5 mM benzamidine, 20 mM N-amidinopiperidine, 0.3 mM benzamidine derivative or 0.6 mM N-amidinopiperidine derivative were titrated into 125, 125, 30, or 60 μM trypsin, respectively. Protein samples were prepared by dialyzing a solution of bovine β-trypsin (Sigma Aldrich, product number T8003) against a buffer composed of 80 mM NaCl, 10 mM $CaCl_2$, 0.1 % (w/v) PEG 8000 and 100 mM HEPES, Tricine or Tris at a pH of 7.6 and a temperature of 4 °C while ligands were dissolved in buffer. In order to support solubility, the buffer additionally contained 3% (v/v) DMSO in the case of peptidomimetic ligands. Titrations were performed in three different buffer systems to enable the correction for any proton transfer occurring between buffer and protein upon complex formation. For this purpose, the obtained ΔH° values were plotted against the ionization enthalpy of the used buffer yielding a buffer-corrected enthalpy as the intercept (for more details see footnote b in Supplementary Tables 2 and 3). Due to small heats-of-dilution generated at the end of each titration, no explicit correction for dilution effects by evaluation of separate ligand-into-buffer titrations was required. Applying the same protocol and data analysis scheme as used in an earlier study[14], ITC titrations were performed on an $ITC_{200}$ device (GE Healthcare) at 25 °C with a reference power of 5 μcal s$^{-1}$. The ITC protocol consisted of an initial 0.3 μl volume injection and of 20–25 injections with a volume between 1.5 and 1.8 μl separated by a 220 s time interval. For data analysis, the method described by Turnbull et al. was applied[45]. The stoichiometry had only to be fixed for N-amidinopiperidine due to its low affinity. This is recommended for titrations with such low c-values[45].

In addition to the above-described ITC experiments, the orthogonal displacement titration protocol was applied for the low-affinity ligands N-amidinopiperidine and benzamidine in the HEPES buffer system to confirm the binding constants determined via direct low c-value titrations. Sample preparations, measurements and data analyses were performed using the same approach as for our earlier study[14], which is described in the following. Initially, the displacement ligand (compound **2**) was characterized by ITC measurements as described above. For displacement titrations, 40 μM trypsin was pre-incubated with 0.17 mM benzamidine or 0.99 mM N-amidinopiperidine, respectively. As in the case of the direct titrations, ligand heat-of-dilutions were minor and thus had not to be explicitly corrected via a ligand-into-buffer control titration. ITC data were collected using an $ITC_{200}$ device (GE Healthcare) at 25 °C with a reference power of 5 μcal s$^{-1}$. An initial 0.3 μl volume injection was followed by 19 injections with a volume of 1.9 μl.

All ITC measurements were performed at least in triplicate and results are presented as mean ± standard deviation (Table 1).

**Quantum-chemical calculations**. DFT calculations of N-amidinopiperidine in its pyramidal conformation were performed with CP2K using the Gaussian plane wave method[46]. Double ζ valence basis sets with one polarization function and Goedecker-Teter-Hutter BLYP pseudopotentials were used for all elements present in N-amidinopiperidine[47,48]. Moreover, we used an energy cutoff of 400 Rydberg for the expansion of the charge density and the DFT-D3 correction to model van der Waals forces[49]. In order to analyze whether and in which situations N-amidinopiperidine is stable in a pyramidal geometry surrounding the piperidine nitrogen, we conducted unrestrained geometry optimizations up to a maximum force of $2 × 10^{-4}$ Bohr$^{-1}$ · Hartree for the pyramidal conformation of N-amidinopiperidine as derived from the corresponding room-temperature X-ray structure in the absence and presence of Asp189.

For trypsin in its complexes with N-amidinopiperidine and benzamidine, short QM/MM simulations have been performed using our previously described approach[14]. In brief, the room-temperature X-ray structures of both protein-ligand complexes were used as starting points after protonation guided by the obtained neutron data (for details see ref. [14] or below in the MD section). In the case of N-amidinopiperidine, two separate simulations were performed starting from each of the two experimentally observed N-amidinopiperidine conformations (see below). During the simulation, the inhibitor, Asp189, Ser190, Gly219, Ala221 and

all water molecules that were located closer than 8 Å from the two Asp189 carboxylate oxygens were included in the QM part. Applying the CP2K code with double ζ valence basis sets (one polarization function) and Goedecker-Teter-Hutter BLYP pseudo-potentials, QM/MM trajectories were generated over 8.4, 7.3 and 8.5 ps for trypsin in its complexes with benzamidine, the planar conformation of N-amidinopiperidine and the pyramidal form of N-amidinopiperidine, respectively. Using the same setup, short in vacuo quantum-mechanical MD simulations were additionally performed in the case of N-amidinopiperidine in both conformations (5.6 and 5.0 ps).

**MD and metadynamics simulations**. The room-temperature X-ray structures of trypsin in its uncomplexed form as well as in complex with benzamidine and N-amidinopiperidine, respectively, were protonated by means of the H++ server (http://biophysics.cs.vt.edu/ H++) at the pH of 7.5 used in the crystallographic experiments[50]. The orientations of rotatable hydrogens were manually adapted based on a comparison with the respective neutron structure. We followed this protocol in order to create a fully protonated model suitable for MD, metadynamics and QM/MM simulations. Hydrogens of this model were placed according to our neutron data or, if not visible in the neutron structures (and only then), as predicted by H++. All simulations were performed with the catalytic His57 in its charged form because the neutron data provided evidence that the His57 δ-nitrogen is fully and the ε-nitrogen partially protonated. This was supported by the H++ program indicating double protonation at pH 7.5. Well-ordered water molecules with $B$-factors below 35 $\text{Å}^2$ and the $Ca^{2+}$ cation were maintained in each starting structure. The AMBER ff14SB force field was used to parameterize trypsin while the ligands' parameters were derived from the generalized AMBER force field (GAFF) except for the amidino group torsion angle relative to the phenyl/piperidinyl moiety[51,52]. Parameters for these torsion angles were generated via DFT calculations (restrained geometry optimizations) applied as described above. For benzamidine, the torsion angle was found to display energetic minima at 40° and 140°. In good agreement with previous calculations[53,54], maxima lie at 0° and 90° representing barriers of 9.5 and 16.0 kJ mol$^{-1}$ height. In the case of N-amidino-piperidine, minima were detected at 20° and 160° while maxima were also found at 0° and 90° with barriers of 2.5 and 38 kJ mol$^{-1}$, respectively. The restrained electrostatic potential method (RESP) was used as implemented in the PyRED server (setting: Gaussian09_E.01) to derive charges for the two trypsin inhibitors[55].

MD simulations of 0.50 and 0.52 μs length were run for uncomplexed trypsin and the trypsin:N-amidinopiperidine complex using the Gromacs software (vs. 4.6.5) on a CPU cluster[56]. Water molecules were treated using the TIP3P model. A dodecahedral box with periodic boundary conditions had been constructed around the protein in a way that each atom of the model and the borders of the cell were separated by at least 12 Å. Neutralization of the system was achieved by the addition of chloride ions. A steepest descent energy minimization over 500 steps was followed by an equilibration in the NPT-ensemble. In this phase, the system was heated from 0 to 297 K over 200 ps and then maintained at 297 K for further 200 ps. During the subsequent production run, NPT conditions and 2 fs time steps were used. Coordinates were saved every 1 ps for uncomplexed trypsin and every 10 ps for the trypsin:N-amidinopiperidine complex. Electrostatics were treated according to the particle-mesh Ewald method with a 10 Å Coulomb distance cutoff. The cutoff for van der Waals interactions was similarly set to 10 Å. A temperature coupling algorithm was used that rescales the velocity with a stochastic term and is similar to the Berendsen thermostat[57]. The reference temperature was chosen as 297 K and the coupling time constant as 0.1 ps. In addition, Parrinello-Rahman pressure coupling with a time constant of 2 ps and a reference pressure of 1 bar was applied.

Metadynamics trajectories of 0.48 and 0.25 μs length were generated for trypsin:N-amidinopiperidine and trypsin:benzamidine using the same basic setup as mentioned above. Instead of CPUs, the runs were performed on GPUs with the program Gromacs (vs. 4.6.1) in combination with Plumed (vs. 2.0)[58]. Coordinates were written every 1 ps. When the distance between the center-of-mass of the ligands' atoms and the center-of-mass of all atoms of Val213, Cys220, Asp189, Ser190, Cys191, Gln192, Asp194, Ser195, Tyr227, His57, Leu99 plus the main chain atoms of Ser214, Trp215, Gly216, Ser217, Gly219, Ala221, Gly193, Pro225, and Gly226 representing the S₁ pocket grew larger than 16 Å, a restraining potential was switched on during the simulation. The resulting confinement region, in which the ligand was allowed to stay, is shown exemplarily for N-amidinopiperidine in Supplementary Figure 6A. The metadynamics-specific settings were 2.5 kJ mol$^{-1}$ for the initial height of the Gaussian hills, Gaussian widths of 0.5 Å and 0.05 rad for collective variable CV1 and CV2, a Gaussian deposition time interval of 2 ps, a temperature of 297 K and a bias factor of 10. Backbone rmsd values were shown to be stable over the simulations (Supplementary Figure 6D and 7B). Both obtained metadynamics trajectories are considered sufficiently converged. Within the final 25 ns of simulation the mean standard deviation of all energy values in the 3D free-energy surface (FES) is 0.57 kJ mol$^{-1}$ for trypsin:benzamidine and 0.29 kJ mol$^{-1}$ for trypsin:N-amidinopiperidine. For the absolute energetic minimum, the standard deviations are even lower with 0.01 and 0.05 kJ mol$^{-1}$, respectively. These values were calculated using 50 data points for each energy value, which were successively generated after 0.5 ns of simulation had been incrementally added until the full trajectory length had been reached. All trajectories were analyzed using Plumed, Gromacs and VMD[59]. Representative snapshots corresponding to

FES minima were selected on the basis of a cluster analysis performed using the method described by Daura et al. with a cutoff of 1.2 Å[60].

**Lifetimes of states of the W1 water molecules**. In this part of the study, all numerical manipulations were carried out using the packages NumPy and SciPy[61]. The Cartesian coordinates were extracted from the trajectory files with the MDTraj package[62]. Initially, the coordinates of water molecules were defined with respect to an internal reference frame, which had been constructed with respect to the position and orientation of individual amino acid side chains in the protein. The principal vectors used for defining each reference frame were build from rigid substructures assigned to specific moieties of the considered side chain. For the amino acids investigated in this study (Asp189, Tyr228, and His57) the axis vectors were defined by the subsets of the side chain atoms listed in Supplementary Table 11. Herein, the z-axis represents the mean orientation of the normal vector of all possible planes that can be constructed from all distinguishable combinations of the z-axis atom subset. The x-axis was defined parallel to the vector connecting the atoms of the x-axis subset. The origin of the coordinate system was defined as geometric center of all atoms in the z-axis and x-axis subsets. By convention, the z-axis was constrained such that its angle with the vector connecting the origin and the geometric center of a neighboring reference amino acid side chain fulfilled the condition $-90° < \alpha < 90°$ (see Supplementary Table 11, 4th column). This ensures that the orientation of the XY plane remains uneffected with respect to the rest of the protein (e.g. the orientation of the internal coordinate system of Asp189 is invariant under rotation of the $C_\beta$-$C_\gamma$ bond).

A water molecule was defined to be bound to an amino acid if its oxygen atom was located within a predefined cutoff of the origin of the reference frame. The cutoff distance was chosen such that it defines the first layer of hydration for the amino acid side chain atom subset used for the definition of the reference frame. The optimal distance was determined from an MD simulation (50 ns) of the respective capped amino acids NME (N-methyl amide capped) and ACE (Acetyl capped) in solution under the same conditions as the MD simulation of the protein. The cutoff distance was then identified as the first minimum in the solvent radial distribution function around the atom subset defining the reference frame. From these MD simulations also solute-solvent interaction properties of the bare, solvent-exposed amino acid were determined and provided a point of reference by representing the maximum of solvent exposure for a particular amino acid.

After the reference frame coordinate system was defined, the Cartesian coordinates of the oxygen and hydrogen atoms of these bound water molecules were transformed to the reference frame coordinate system. Furthermore, the rotational coordinate space spanned by the water molecules was described by applying an internal coordinate system to each water molecule within the hydration layer cutoffs: The O atom was defined as the origin of the water internal coordinate system. Then, the x-axis, $R_x$, was set to be the O–H1 bond vector and the z-axis, $R_z$, was defined as being orthogonal to the O–H1 and the O–H2 bond vectors. Finally, the y-axis, $R_y$, was obtained as the cross product of x and z-axes vectors. Please note that the water internal coordinate system was constructed from the Cartesian coordinates of the water molecules in the respective frame of reference as outlined above.

The lifetime of a particular state (with respect to the translational or rotational degrees of freedom) that a specific water molecule can occupy was defined as the time span the water molecule is present in that state. For the sake of simplicity, only a single translational state was defined. This state was considered occupied when the oxygen atom of a water molecule was within the cutoff distance to the origin of the reference frame of the amino acid. In that manner, for each water molecule $j$ that occupies the site, a survival function $B_j$ was defined, that is summed over a number of $N_f$ frames, where $N_f$ is the number of frames along the entire or a chunk of the MD trajectory. This function was assigned 1 at position $t'$ if the state is occupied during the $t'$ th frame and the $t + t'$ th frame of the trajectory and 0 otherwise. Since only a single continuous series of occupied states is of interest for the calculation of the lifetime, the state must not be unoccupied in between frames $t'$ and $t + t'$. Consequently, the following pseudo-autocorrelation functional, which runs over all water molecules, $N_W$ that occupy the site, was defined for the purpose of translational states:

$$C_{\text{trans}}(t) = \sum_{j=0}^{N_W} \sum_{t'=0}^{N_f-t} B_j(t') \prod_{k=t'}^{t'+t} B_j(k) \qquad (1)$$

The lifetime for the translational state $\tau_{\text{trans}}$ was obtained by normalizing $C_{\text{trans}}(t)$ with respect to $C_{\text{trans}}(0)$ and integrating as follows:

$$\tau_{\text{trans}} = \int_0^\infty C_{\text{trans}}(t)\,dt \qquad (2)$$

The evaluation of Eqs. 1 and 2 was separately performed over chunks of 2 ns in length and then averaged. The actual integration was carried out using the trapezoidal integration scheme as implemented in NumPy.

The lifetimes of rotational states of water molecules were only calculated for those water molecules that occupy the translational state as described above. Although it is straightforward to describe an internal coordinate system for rigid water molecule models, the assignment of meaningful rotational states is not unambiguous. We therefore investigated the rotational diffusion of the individual coordinate axis vectors of the water molecule internal coordinate system, which were defined with respect to the reference frame of the amino acid as explained above. The time evolution of these was determined by superimposing the autocorrelation function (Eq. 1) with the autocorrelation of the dot product of the individual coordinate axis vectors in the water coordinate system:

$$C_{\mathrm{rot}}(t) = \sum_{j=0}^{N_W} \sum_{t'=0}^{N_f-t} \left( R_j(t') \cdot R_j(t'+t) \right) B_j(t') \prod_{k=t'}^{t'+t} B_j(k) \qquad (3)$$

In Eq. 3, $R$ is one of the coordinate axis vectors $R_x$, $R_y$, or $R_z$ in the water coordinate system and $C_{\mathrm{rot},x}$, $C_{\mathrm{rot},y}$, or $C_{\mathrm{rot},z}$ are the corresponding autocorrelations. The different rotational lifetimes $\tau_{\mathrm{rot},x}$, $\tau_{\mathrm{rot},y}$, and $\tau_{\mathrm{rot},z}$ were then calculated using eq. 2.

**Code availability**. Our previously published automated refinement pipeline script mentioned in the above experimental section is available from the corresponding author upon request[20].

## Data availability

The crystallographic datasets and structures generated during the current study are available in the Protein Data Bank (PDB) repository with the accession codes 5MNZ (apo trypsin neutron structure), 5MO2 (trypsin:N-amidinopiperidine neutron structure), 5MO0 (trypsin:benzamidine neutron structure), 5MNE (apo trypsin X-ray structure at 100 K), 5MNN (trypsin:N-amidinopiperidine X-ray structure at 100 K), 5MNG (trypsin: benzamidine X-ray structure at 100 K), 5MNQ (trypsin:D-Phe-Pro-N-amidinopiperidine X-ray structure at 100 K), 5MNF (apo trypsin X-ray structure at 295 K), 5MNO (trypsin: N-amidinopiperidine X-ray structure at 295 K), 5MNH (trypsin:benzamidine X-ray structure at 295 K), 5MOP (apo trypsin XN structure), 5MOS (trypsin:N-amidinopi peridine XN structure), and 5MOQ (trypsin:benzamidine XN structure). All other generated data are included in this published article and its supplementary information files or are available from the corresponding author upon request.

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

## Acknowledgements

We thank the Heinz Maier-Leibnitz Zentrum (Munich, Germany) for generous allocation of beamtime at the FRM II and for travel support. In addition, we are grateful to the staff of the EMBL beamlines at the DESY for their support during the collection of ultra-high resolution and room temperature X-ray data. We also thank the Helmholtz-Zentrum Berlin for travel support and their beamline staff for assistance. We also thank Pavel Afonine, Chad Brautigam and Andre Mitschler, who supported our work with advice and technical help concerning refinement, ITC evaluation and the mounting of large protein crystals in capillaries. We are grateful to Hans-Dieter Gerber, Benjamin Wenzel, Torsten Steinmetzer, Anna Sandner and Namir Abazi for supporting the qNMR measurements and enzyme inhibition studies. Moreover, we are thankful for the financial support of J.S. by a short-term fellowship granted by EMBO. The presented work was supported by the European Research Council (ERC) of the European Union (grant 268145-DrugProfilBind) awarded to G.K.

## Author contributions

Crystallographic work was performed by J.S., A.O., T.E.S., and A.H. Computational studies were conducted by J.S., R.G., T.W., and A.C. ITC measurements were performed by J.S. Inhibition data were collected by J.S. and C.S. The D-Phe-Pro analog of N-amidinopiperidine was synthesized by K.N. The research was designed by A.C., A.H., and G.K. with contributions from J.S. and R.G. The manuscript was written by J.S., G.K., and A.H. with contributions from all authors.

## Additional information

**Competing interests:** The authors declare no competing interests.

