## [Peer Review File · Nature Communications]

Reviewers' comments:

Reviewer #1 (Remarks to the Author):

The study describes three neutron structures of trypsin in its apo form and bound to two different inhibitors. The exceptionally high resolution obtained for these neutron structures allow a proper orientation of water molecules in both the uncomplexed and complexed states of trypsin. In the uncomplexed state, water molecules in the S1 pocket, which have a reduced number of H-bonds, are less stable than water molecules from the reservoir. This facilitates their removal from the S1 pocket, needed during the binding process of the studied inhibitors. To further understand the role of water during the binding and unbinding process, metadynamics have been performed, revealing the choreography of water molecules during the (un-)binding events of ligands.

While all the results presented in this study are solid, the authors fail to clearly present the point claimed at the end of the abstract: "reconsidering our current view on water influencing protein-ligand recognition". Indeed, the results do not present how important they are in the field, and what is there additional value in the role of water in protein-ligand recognition. Therefore, I do not recommend publication of this article in Nature communications.

Suggestion to improve manuscript:

The title is somehow misleading: most of the conclusions drawn on the role of water, come from the simulations.

Also, the manuscript should present a stronger link between the simulations and the neutron structures. For example, the calculations made on the water molecule W1 of the pocket S1, could have also been made on W2 and W3, and compared to water molecules from the reservoir to clearly show the difference in the dynamical behaviour between the reservoir and the S1 pocket. The complete paragraph on water molecule W1 should be moved from the supplementary to the main text.

Also, after an unbinding event observed in the simulation, does the water molecules fill the S1 pocket as in the uncomplexed state, and have comparable dynamics ?

Also, the complete paragraph on the planar and pyramidal geometries of the N-amidinopiperidin should be moved to the supplementary (as well as corresponding figures). It distracts the reader from the main point of the manuscript.

The introduction is rather succinct, and does not properly introduce how the presented results are of relevance in the comprehension of the role of water in ligand binding processes. I would also revise the statement concerning the necessity of millimetre long crystal on edge. If you mention crystal sizes, or challenging techniques, it would be fair to mention other limitations (few neutron instruments if compared to X-ray) or improvement (deuteration helps to lower the crystal size). The conclusion/discussion suffers from the same pitfall.

Aso, all structural comparisons with thrombin should be removed from the manuscript if the thrombin structure mentioned is not available. (ref17 mention this structure but does not describe it).

Minor comments:

P9: please provide values of Mulliken charges, so that the reader can estimate what the slight increase represent.

P11: It is stated that modelisation of W3 in the complex state is impossible, while explicit residual density is visible. Does-it mean W3 was placed in the model, and its occupancy refined, and it reached an occupancy of 0 ? Actually, what does "impossible" mean here ?

P12 - lines 245 - 249. This long sentence is not easy to read and could be rewritten or split in two.

P18: the enzyme understood to feature its S1 pocket...

Please rephrase.

Reviewer #2 (Remarks to the Author):

The authors present a very complete structural (combining X-ray and neutron data) and computational study, carefully discussing the role of water in the ligand binding events in the example of trypsin. Through the combination of the different crystal structure information and MD simulations, they deliver a sound picture of the influence differential solvation can have in the affinity of ligands, and the barriers to binding.

I believe their conclusions are well justified and only have one major comment, which needs to be resolved and a few minor comments. After these are taken into consideration, I would wholeheartedly recommend publication.

major comment: The description of the QM/MM calculations is completely missing. There is no description of what is included in the QM region, what QM method was used, what QM program was used,... etc... I was unable to find any details in the SI. Also the QM calculations are not properly documented (for example, to calculate the results in Fig. 4 or to carry out optimisations). What level of theory was used? Simply stating DFT is far from being enough.

minor comments:

1- I fail to understand the results in Fig. 4 for the complex with aspartate. In the caption it is only mentioned constrained DFT optimisations. In the text there is a mention to QM/MM results. Which one was it? Also, the profile for the complexed state seems highly irregular. If this is the result of a QM/MM estimate (time averaged) some fluctuation could be expected, but the jagged pattern looks very suspicious.

2- again Fig. 4. There is no value for the harmonic constraint. Also, when there is an harmonic constraint, the real value angle will differ from that set in the constraint as ideal. Usually, it is better to show the real value, as only then the result is independent of the constraint.

3- and again to Fig. 4. The scale of energies shown makes little sense. It would be best to set a reference point and plot the energies relative to that value. This would also help understand the difference in stiffness between the two cases (free vs. complexed) more easily.

Reviewer #3 (Remarks to the Author):

The paper describes a thorough analysis of the protein-ligand recognition with the focus on water molecules playing a fundamental role in this process. The chosen combination of several methods resulted in a very interesting, well done and an in-depths study. I can only hope that this paper will encourage other researchers to try to push the highest resolution limit even further and combine similar analysis with an ultra-high resolution crystal structure and charge density analysis. The presented results complete our general view of ligand recognition process and thus facilitate a more reliable design of novel drugs. This manuscript merits publication in Nature Communications.

I have only a few questions, and suggestions for minor corrections:

1. The authors mention refinement using the SHELXL program which was most likely the full-matrix refinement performed in order to estimate deviations in the local geometry surrounding the endocyclic guanidino nitrogen. One can assume that the deposited model was refined using another program (PHENIX, REFMAC5). If so, this should be clearly stated. Although the observed geometrical differences are undeniable, some details describing the utilized refinement procedure could be of interest for potential readers. More specifically: Which atoms were geometrically unrestrained during the refinement? Were hydrogens of the protein added by the SHELXL-implemented 'riding model'? Did you use enhanced rigid-bond restraints during the refinement? SHELXL citation is only present in SI.

2. What was the reason to truncate the highest resolution limit of X-ray diffraction data? In particular, relaxing the "old" rules could significantly increase the highest resolution limit for D-Phe-Pro-N-amidinopiperidine, e.g.: $CC(1/2) > 0.5$, $I/\sigma(I) > 1.5$, Rmerge for the highest resolution shell $< 70\%$. CC1/2 statistics is not included.

3. Isotopes are not normally considered to be distinguishable by X-ray crystal structure analysis. However, it has been suggested that the difference in the vibrational behavior of deuterium and protium could provide a way to distinguish these two isotopic species by X-ray diffraction. Could you confirm it based on your experimental data?

4. PROCHECK is a bit outdated program for checking the stereochemical quality of a protein structure. In order to conform to "current standards" the stereochemical quality of a protein should be validated using Molprobity program.

A few minor comments.

Page 3. The authors state that "X-ray crystallography is unable to detect hydrogen atoms reliably". Although the next sentence provides a more detailed description concerning the usual resolution range achieved by that experimental technique, the statement it is not fully correct. Certainly, detection of hydrogens depends on several parameters, e.g.: the quality of crystal, the highest resolution limit, completeness of diffraction data, ADP, disorder/occupancy, etc. Certainly, not all hydrogen atoms can be detected based on the electron density map. However, X-ray crystallography has been successfully used to study enzymes with bound substrates/products at true atomic resolution. Although not all positions of hydrogen atoms could be assigned unambiguously on the basis of the electron-density maps, the protonation state of same, important for the mechanism, groups could be deduced.

Page 6. "certain protein functionalities" sounds a bit odd to me. It could be replaced by "certain protein functional groups such as hydroxyl moiety"

Page 10. "two scattered configurations" could be described as "two distinguished configurations".

Page 10. "Apart from this interaction, it appears difficult for W1 to find an additional partner for an optimal interaction" sounds personal. Describing it as the lack of additional interaction partners is more neutral.

Page 11. The description "this water experiences enhanced residual mobility, it clearly sacrifices rotational degrees of freedom upon ligand accommodation" is not wrong, just a bit personalizing W1 water molecule. :-)

Reviewer #4 (Remarks to the Author):

Schiebel et al investigates the role of water molecules in the S1 binding pocket in the apo and ligand bound forms of bovine trypsin using X-ray and neutron crystallography. The data appears to be high-resolution and very good quality. I have some concerns about the derived crystallographic models and the conclusions of this study.

Methods:

Crystallography:

The flow of crystallographic models and Rfree flags of reflection have to be better described. Where the starting structure of trypsin was coming from? How were the Rfree flags generated and are they related between the different structures? In most of the structures the gap between Rfree and R is not that great (<1%), suggesting that the Rfree sets were de novo generated and a limited rounds of (automated) refinement were performed. At the same time there is a great variation in Rfree between closely related structures suggesting different origins of Rfree sets (11% vs 17%). For very high resolution X-ray structures allocating 5% Rfree flags is not motivated on statistical grounds (2000-3000 reflections are typically sufficient) and weakens the experimental evidence. Since the paper is focusing very much on the S1 site it is unclear how much scrutiny the rest of the structure received. Could the evidence substantially alter the crystallographic models during automated refinement? I am reasonably sure that the S1 pocket was analyzed carefully, but there are other, arguably more important, regions such as the catalytic triad where the position, presence or absence of a hydrogen/deuterium has very important implications. In ultra-high resolution structures there is substantial evidence for disorder. It is necessary to describe how these were modelled, how many residues (and which atoms) were treated as alternative conformers in each structure. Figures showing the model and electron density in these regions are also necessary to validate the model building and refinement strategy.

MD simulation:

I appreciate the efforts of the authors to manually curate the hydrogen positions guided by the neutron diffraction data. I think it is reasonable to fill missing positions for non-titrable hydrogens by reasonable prior knowledge, but for titratable hydrogens the experimental evidence must take precedence. Motivation like we protonated the epsilon N of His-57, because some continuum electrostatics method determined so in my view is not acceptable. Otherwise what is the point of doing neutron diffraction experiments? In my opinion, the protonation state of His-57 at pH 7.5 deserve better analysis than hiding it in the supplementary methods section. If for no other reason, because of its close proximity to the S1 binding site.

Conclusion:

It is safe to say that a conclusion with more than 1000 words is excessive and at times it reads like pure discussion/speculation. For example uncharged inhibitors, regulatory sodium ions and comparison of position 192 in trypsin and thrombin are only mentioned in the conclusion without any support in previous sections. I suggest the authors condense their main conclusion into focused points like in their earlier *Angewandte Chemie* paper.

Minor points:

Introduction

"Unfortunately, not much experimental data have been collected deciphering how water molecules act exactly during protein-ligand complex formation."

Bulk and interface water was successfully studied with far-IR and optical Kerr effect spectroscopy, I suggest the authors extend their literature study, because experimental data is available.

"The situation improves significantly once the resolution gets better than 2.0 Å with distinct water orientations discernible at 1.5 Å or better."⁷

Neutron diffraction is also an averaging technique and thus affected by local disorder even at very high resolution. The high-resolution structures presented here do not have 100% detection of hydrogen positions.

Page 17 "Furthermore, the chloro-substituted ligands establish an electrostatic and thus

enthalpically favored chloro-aromatic contact"

London dispersion forces are better explanation for the stabilization of chloro-aromatic contact.

Reviewer #1 (Remarks to the Author):

The study describes three neutron structures of trypsin in its apo form and bound to two different inhibitors. The exceptionally high resolution obtained for these neutron structures allow a proper orientation of water molecules in both the uncomplexed and complexed states of trypsin. In the uncomplexed state, water molecules in the S1 pocket, which have a reduced number of H-bonds, are less stable than water molecules from the reservoir. This facilitates their removal from the S1 pocket, needed during the binding process of the studied inhibitors. To further understand the role of water during the binding and unbinding process, metadynamics have been performed, revealing the choreography of water molecules during the (un-)binding events of ligands.

While all the results presented in this study are solid, the authors fail to clearly present the point claimed at the end of the abstract: "reconsidering our current view on water influencing protein-ligand recognition". Indeed, the results do not present how important they are in the field, and what is there additional value in the role of water in protein-ligand recognition. Therefore, I do not recommend publication of this article in Nature communications.

We believe that the aspect of the detailed water structure prior to ligand binding is an essential part of protein-ligand recognition, as the binding event is an inventory of the state **before** and **after** binding. Often enough, the situation of the state before binding is not considered or even unknown. In case of X-ray diffraction, the proper water pattern is usually impossible to resolve sufficiently, and no insights from an experimental point of view are given on the rotational and translational states of water molecules in the different states. In our series of detailed experimentally resolved structures, we provide important evidence on this inventory. We show the imperfect hydration of Asp189, which is supposedly the prerequisite for potent substrate and inhibitor binding to trypsin-like serine proteases. Furthermore, the change in ordering states of the water molecules, which mediate contacts between protein and ligand are observed which correlate with binding thermodynamics. We believe these aspects are novel and will improve our understanding of the ligand-binding inventory. Some ideas might have been anticipated before *but* only on a conceptual level and not on a sound experimental basis as we provide it here in this study to strengthen some of these key hypotheses. We therefore regret that the reviewer concludes that our "*results do not present how important they (water influence) are in the field, and what is their additional value in the role of water in protein-ligand recognition*". In the revised version of the manuscript, we tried our best to further emphasize why our results are important for a better understanding of the ligand-binding process. In this regard, we thank the reviewer for his below suggestions for an improvement of the manuscript. Please also note that reviewers 2 and 3 clearly recognize the value and importance of the current work when they write that the "*presented results complete our general view of ligand recognition process and thus facilitate a more reliable design of novel drugs*" and that the "*paper describes a thorough analysis of the protein-ligand recognition with the focus on water molecules playing a fundamental role in this process*".

Suggestion to improve manuscript:

The title is somehow misleading: most of the conclusions drawn on the role of water, come from the simulations.

We think that the most important novel insights are actually provided by the neutron diffraction data, which has been lacking before to conclusively show how important waters are and how they change their properties in the ligand-recognition process. The evidence in the experimental data comprises the H-bonding geometry, water ordering states and configurational differences due to hybridization.

Furthermore, we recorded the thermodynamic binding data experimentally. Nevertheless, we agree with the reviewer that we need the simulations to show that, e.g., the crystallographic disorder correlates with differences in the **dynamic** properties (and not a **static** disorder!) or the charge distributions on the molecules that are important for the binding differences. Thus, in summary, we think that our conclusions are primarily based on experimental evidence and **not** computation. However, the simulations are needed to provide an explanation why they are observed in the experiments. A title has to be informative to attract putative readers, we therefore suggest to keep the key word "Neutron Crystallography" in the title.

Also, the manuscript should present a stronger link between the simulations and the neutron structures. For example, the calculations made on the water molecule W1 of the pocket S1, could have also been made on W2 and W3, and compared to water molecules from the reservoir to clearly show the difference in the dynamical behavior between the reservoir and the S1 pocket. **The complete paragraph on water molecule W1 should be moved from the supplementary to the main text.**

We have made additional calculations that attempt comparing both, the simulations and neutron structures on the structural level. A short description of the results can be found in the main manuscript and details on the methodology in the Experimental Section. Briefly, the crystallographic W1 water molecules are found to differ a bit from the MD simulation (on average 2.82 Å between the crystallographic water oxygen and its first nearest neighbor in each MD frame). As also observed in the crystallographic experiment, both orientations of W1 water molecules are equally well reproduced (hydrogen-hydrogen RMSD_{H-H} 1.04 and 0.91 Å for conformation A and B, respectively). Remarkably, the position of W3 is also well reproduced, within 1.12 Å of the crystallographically detected position, which is even better than the position of structurally-bound water molecule W7.

In order to further probe the dynamic behavior of solvation pattern in the S1 pocket, we compared the dynamic properties of the water molecules in the first hydration layer of Asp189 (corresponding to W2 and W3) and Tyr228 (corresponding to W1) with respect to the corresponding one found for water molecules of the bare (capped) amino acids in pure solvent. With this type of analysis, the contributions of the individual amino acids can be separated from the effect imposed by the protein environment. Accordingly, individual damping factors were defined, which assess the ratio of the residence times of the considered water molecules (e.g. translational residence time, τ_{trans}) bound to the amino-acid site in the protein and to the corresponding site in pure solvent. These factors should be much larger than 1, if the local dynamics are damped. If they become closer to 1, the dynamic behavior of the water molecules in the first hydration layer are hardly influenced by the protein environment, which we expected for solvent-exposed residues.

The results of these rather elaborate, additional investigations performed during the revision phase are now presented in the Experimental Section and as Supporting Notes and in essence, they are summarized in the main manuscript. Briefly, water molecules W2 and W3 are dynamically strongly confined within the first hydration layer of Asp189, as indicated by their damping factors ranging from 14 to 24. However, W1, which sits on top of Tyr228, experiences a reduced damping, as indicated by factors ranging from 5.7 to 7.1. This observation suggests that the protein environment takes stronger impact on the local dynamics of the solvation properties next to Asp189 compared to Tyr228. Additional conclusions, drawn from these findings, are summarized in the main manuscript.

Also, after an unbinding event observed in the simulation, does the water molecules fill the S1 pocket as in the uncomplexed state, and have comparable dynamics?

We apologize that we seem not to have made this clear enough in the first version of the manuscript. Absolutely, water molecules flood the S1 pocket after the ligand *N*-amidinopiperidine (or benzamidine) has left the site. This is clearly visible in Movies S1 and S3 toward their end. We tried to better communicate this important aspect to the reader in that we complemented the following sentence of the manuscript and included a reference to the movies, which unfortunately had been missing before: "*Subsequently, additional water molecules penetrate into the binding site most frequently via the main entrance of the S₁ pocket, ultimately leading to the fully dissociated state and a hydrated ligand-free S₁ pocket (Movies S1 and S3).*" The water structure prior to a binding event and following an unbinding event in the metadynamics simulation is highly similar as recorded in the trajectories and upon comparison of Movies S1 and S3 (unbinding events) with Movie S2 (binding event) although this is not the main purpose of these movies. With regard to the dynamics of these waters during the metadynamics simulation, we would like to stick to the phenomenological description and refrain from a detailed structural analysis since this is beyond the scope of this type of simulation which requires a bias factor to enable a simulation of multiple (un)binding processes in the accessible time span and thereby the reconstruction of the free-energy landscape. Actually, this is why we chose to additionally investigate the ligand-free state in a long and completely unbiased simulation. In the revised manuscript, we extended the analysis of the latter simulation (see comment above) and now more clearly provide the link to the metadynamics simulations. In summary, the metadynamics simulations show that a comparable water structure exists prior to ligand binding and following ligand unbinding, which is in accordance with the kinetic principle of micro-reversibility. The ligand-free state itself was then more closely investigated in a pure MD simulation of apo trypsin, profiting from the extremely well defined structural knowledge gained through our crystallographic study.

Also, the complete paragraph on the planar and pyramidal geometries of the *N*-amidinopiperidine should be moved to the supplementary (as well as corresponding figures). It distracts the reader from the main point of the manuscript.

While the discussion on the planar/pyramidal geometry of the *N*-amidinopiperidine might not be the main point of this manuscript, we think it is a very important and unique observation only made possible due to the ultra-high resolution X-ray structure. Likely, it would hardly be found by pure simulations and only our experiments triggered the study of its relevance. The experimental findings stimulated us to investigate the surprising pyramidalization further by performing CSD searches and QM calculations. We believe our findings are important because they show the impact of the charge concentration on the Asp189 residue in trypsin-like serine proteases. Please note that this residue is directly involved in substrate recognition and the charge concentration on Asp189 can take influence on substrate specificity. Therefore, we would like to keep this part in the manuscript as is. Additional information on this topic are already given as Supplementary Notes, as we are aware of limitations in manuscript length.

The introduction is rather succinct, and does not properly introduce how the presented results are of relevance in the comprehension of the role of water in ligand binding processes. I would also revise the statement concerning the necessity of millimetre long crystal on edge. If you mention crystal sizes, or challenging techniques, it would be fair to mention other limitations (few neutron instruments if compared to X-ray) or improvement (deuteration helps to lower the crystal size). The conclusion/discussion suffers from the same pitfall.

We thank the reviewer for the hint that the introduction and conclusion/discussion sections might benefit from additional emphasis on what the study contributes to what was known previously from available literature. We have modified these parts accordingly. Concerning the other statement, the referee is of course right that in neutron diffraction there are additional limitations such as fewer neutron than X-ray instruments and comparatively low flux. Also deuteration is beneficial, albeit

crystals of deuterated material tend to be smaller in size *per se*. Examples of data collection with such crystals have required very long data collection times (up to 93 days/dataset see for example M.P. Blakeley et al., Quantum model of catalysis based on a mobile proton revealed by subatomic X-ray and neutron diffraction studies of h-aldose reductase, PNAS 105, 2008, 1844 -1848). While the experimenter has no influence on the number of instruments and flux, the only variable he can control is crystallization conditions and crystal growth. The importance of crystal growth is acknowledged in the literature, where a chapter in the "Neutron Protein Crystallography" book is devoted to this topic (N. Niimura, A. Podjarny; Neutron Protein Crystallography, Oxford University Press 2011, 50-73). We stated in our manuscript "*To some degree this challenge arises from the fact that very large single crystals of millimeter edge length are required.*" Therewith, we acknowledged, that crystal size to some degree is important among other factors, some of which we now mention in the revised version of the text. Nevertheless, we are convinced that the exceedingly good resolutions we obtained for our neutron structures within a reasonable beamtime were only possible because we were able to grow crystals of such size.

Aso, all structural comparisons with thrombin should be removed from the manuscript if the thrombin structure mentioned is not available. (ref17 mention this structure but does not describe it).

In fact, Reference 17 (Rühmann et al., J. Med. Chem. 58, 6960-6971, 2015) in the initially submitted manuscript reports the crystal structure of thrombin in complex with *N*-amidinopiperidine. Nevertheless, we agree with the reviewer that we should have additionally mentioned the PDB code of this structure (4UE7), which is the case in the revised manuscript version.

Minor comments:

P9: please provide values of Mulliken charges, so that the reader can estimate what the slight increase represent.

We apologize that we have not provided the values in the initial manuscript version. They were somewhat hidden in Fig. 2 and now are mentioned explicitly in the text.

P11: It is stated that modelisation of W3 in the complex state is impossible, while explicit residual density is visible. Does-it mean W3 was placed in the model, and its occupancy refined, and it reached an occupancy of 0 ? Actually, what does "impossible" mean here ?

In the manuscript we had written the following: "*Residual density indicates, however, that the former W3 site is still partly occupied albeit it was insufficient to refine W3 in the structural models (Fig. 2A and C).*" In principle, it was possible to refine W3 with an occupancy > 0. Since B-factors and occupancy are intrinsically related and a meaningful occupancy is, thus, difficult to refine for water molecules, we usually refrain from a water occupancy refinement unless the water molecule is coordinated to a partly occupied side chain. When refining the structure, we set a meaningful B-factor cutoff for water molecules and this particular water did not meet our strict criteria, which are based on a discussion of protein structure refinement in "Crystal Structure Refinement, edited by P. Müller, Oxford University Press 2006, page 179f." Accordingly, we also do not include water molecules at the surface that result in lower B-factors but do not interact convincingly with the protein. While we could possibly lower the crystallographic R-values we also consider if water placement is sensible to avoid overfitting by addressing every bit of residual electron density which might be somewhat conservatively, but avoids flooding of structures with meaningless water molecules. In order to make our statement less misleading, we rephrased it as follows: "*Residual density, however, indicates that the former W3 site is still partly occupied albeit we refrained from refining W3 in the structural models due to low*

occupancy or high B-value when assuming full occupancy (Fig. 2A and C)." Thereby, we did not include it in the structure coordinate file, which might be used by anyone without paying close attention to it, but still acknowledge a partial presence by showing and briefly discussing its density in the manuscript.

P12 - lines 245 - 249. This long sentence is not easy to read and could be rewritten or split in two.

We agree with the reviewer and split the sentence in three separate sentences, each of which were further modified so that they are easier to comprehend.

P18: the enzyme understood to feature its S1 pocket...
Please rephrase.

According to the referee's suggestions, we rephrased this part and modified it to "*the S1 pocket of the enzyme evolved in a way...*"

Reviewer #2 (Remarks to the Author):

The authors present a very complete structural (combining X-ray and neutron data) and computational study, carefully discussing the role of water in the ligand binding events in the example of trypsin. Through the combination of the different crystal structure information and MD simulations, they deliver a sound picture of the influence differential solvation can have in the affinity of ligands, and the barriers to binding.

I believe their conclusions are well justified and only have one major comment, which needs to be resolved and a few minor comments. After these are taken into consideration, I would wholeheartedly recommend publication.

major comment: **The description of the QM/MM calculations is completely missing.** There is no description of what is included in the QM region, what QM method was used, what QM program was used,... etc... I was unable to find any details in the SI. Also the QM calculations are not properly documented (for example, to calculate the results in Fig. 4 or to carry out optimisations). What level of theory was used? Simply stating DFT is far from being enough.

We apologize if the information about the QM/MM and QM calculations was not easy to find in the text. In fact, the Supporting Information file did contain most of the information the reviewer is asking for. According to our restructuring of the manuscript, they now can be found on pages 31 and 32 of the Experimental Section on "Quantum-chemical calculations" where the first paragraph describes the QM calculations and the second paragraph the QM/MM calculations. In order to help the reader identifying this important information, **we included all method descriptions in the main text.** Furthermore, we carefully revised the description of the methods and added some more information that had been missing before (e.g. QM method). Please note that the definition of the QM region is clearly described in the QM methodological section: "*During the simulation, the inhibitor, Asp189, Ser190, Gly219, Ala221 and all water molecules that were located closer than 8 Å from the two Asp189 carboxylate oxygens were included in the QM part.*" It is also important to consider that we applied the same QM/MM approach as in our previous study (Schiebel *et al.*, *Angew. Chem. Int. Ed. Engl.* **2017**, *56*, 4887-4890) and therefore referred to the method section of this publication and only briefly repeated the most important facts and minor modifications as stated in the method description: "*For trypsin in*

its complexes with *N*-amidinopiperidine and benzamidine, short QM/MM simulations have been performed using our previously described approach.¹¹

minor comments:

I fail to understand the results in Fig. 4 for the complex with aspartate. In the caption it is only mentioned constrained DFT optimisations. In the text there is a mention to QM/MM results. Which one was it? Also, the profile for the complexed state seems highly irregular. If this is the result of a QM/MM estimate (time averaged) some fluctuation could be expected, but the jagged pattern looks very suspicious.

We agree with the reviewer that Fig. 4 was not well explained in the initial version of our text. In the revised manuscript, we modified the respective parts in the main text and figure caption. Furthermore, we added a reference to the methodological details previously described in the Experimental Section and in the main text (see above). Fig. 4 shows the results from restrained DFT geometry optimizations (for *N*-amidinopiperidine in black and the *N*-amidinopiperidine:aspartate complex in red). The QM/MM results mentioned in the main text, in contrast, are illustrated in Fig. S3. This should now become clear from the revised manuscript where the energy profiles in Fig. 4 are described to originate from "restrained DFT geometry optimizations for free *N*-amidinopiperidine and the *N*-amidinopiperidine:aspartate complex". The restrained DFT geometry optimizations for the complex have been performed analogous to those for the individual *N*-amidinopiperidine molecule. Since a smooth energy profile resulted for *N*-amidinopiperidine, we interpret the jagged energy profile of the complex as a profile with multiple local energy minima. This is in line with the finding that the unrestrained geometry optimization of *N*-amidinopiperidine results in a planar molecule with a torsion of $\sim 0^\circ$ (black dashed vertical line), which approximately coincides with the minimum of the corresponding energy profile (black line) while the complex relaxes only slightly during an unrestrained optimization to the next local minimum (red dashed vertical line and red profile). The minor local minima observed for the *N*-amidinopiperidine:aspartate complex are likely related to a subtle interplay of forces resulting in mechanical stability.

2- again Fig. 4. There is no value for the harmonic constraint. Also, when there is an harmonic constraint, the real value angle will differ from that set in the constraint as ideal. Usually, it is better to show the real value, as only then the result is independent of the constraint.

We added the chosen force constant of the harmonic restraint in the figure caption. This value had been mentioned in the SI but was a bit too hidden. In order to help the reader to better understand the figure, we now also explicitly refer to the methods section (previously in the SI and now in the main text) where the approach is described in more detail. Moreover, we agree with the reviewer that the angle value resulting from the optimization is the much better choice than the target value of the restraint/constraint. This is why we had chosen these real improper torsion angle values for the plot in Fig. 4. We are sorry that this was not mentioned explicitly in the figure caption and explain this now in the revised version of the caption.

3- and again to Fig. 4. The scale of energies shown makes little sense. It would be best to set a reference point and plot the energies relative to that value. This would also help understand the difference in stiffness between the two cases (free vs. complexed) more easily.

Since the resulting energies are only relative values anyhow, we decided to adapt Fig. 4 according to the reviewer's suggestion in order to facilitate interpretation of the figure (only one y-axis required).

As a common reference point, we chose the energy at an improper torsion angle of 0°. This energy was set to 0 kJ/mol in both instances.

Reviewer #3 (Remarks to the Author):

The paper describes a thorough analysis of the protein-ligand recognition with the focus on water molecules playing a fundamental role in this process. The chosen combination of several methods resulted in a very interesting, well done and an in-depths study. I can only hope that this paper will encourage other researchers to try to push the highest resolution limit even further and combine similar analysis with an ultra-high resolution crystal structure and charge density analysis. The presented results complete our general view of ligand recognition process and thus facilitate a more reliable design of novel drugs. This manuscript merits publication in Nature Communications.

I have only a few questions, and suggestions for minor corrections:

1. The authors mention refinement using the SHELXL program which was most likely the full-matrix refinement performed in order to estimate deviations in the local geometry surrounding the endocyclic guanidino nitrogen. One can assume that the deposited model was refined using another program (PHENIX, REFMAC5). If so, this should be clearly stated. Although the observed geometrical differences are undeniable, some details describing the utilized refinement procedure could be of interest for potential readers. More specifically: Which atoms were geometrically unrestrained during the refinement? Were hydrogens of the protein added by the SHELXL-implemented 'riding model'? Did you use enhanced rigid-bond restraints during the refinement? SHELXL citation is only present in SI.

In the Supporting Information (now Experimental Section) we already stated that a least-square blocked matrix refinement in SHELXL was used. Sorry, we used the "BLOC algorithm" statement in the SI, which is program specific, but stands for blocked matrix least-squares. We rephrased accordingly in the revised version of the manuscript (please note that all method descriptions have been copied from the SI to the main text): "*In the case of trypsin:N-amidinopiperidine, not only PHENIX was used to generate a structural model but the corresponding data collected at 100 K were additionally refined against intensities using a blocked matrix least-squares (BLOC) algorithm in SHELXL.*"⁸ This sentence now also more clearly expresses that the deposited structure has indeed been refined using PHENIX in order to be consistent with all other X-ray structures. To make this even clearer to the interested reader, we further introduced the following sentence in the Refinement against X-ray diffraction data section in the methods part of the manuscript: "*Refinements of the models deposited in the PDB have been performed against structure factor amplitudes using PHENIX.*" Also, the SHELXL citation has been added to the main body of text. The description of the refinement itself (restraints, model details, rigid-body restraints) has been extended and has been moved from the SI to the main text.

2. What was the reason to truncate the highest resolution limit of X-ray diffraction data? In particular, relaxing the "old" rules could significantly increase the highest resolution limit for D-Phe-Pro-N-amidinopiperidine, e.g.: $CC(1/2) > 0.5$, $I/\sigma(I) > 1.5$, Rmerge for the highest resolution shell $< 70\%$. $CC1/2$ statistics is not included.

In general, we are more conservative with respect to the high resolution cut-off. For instance we usually apply the widely used $I/\sigma(I)$ criterion > 2.0 explaining our choice of the high resolution cut-off for the D-Phe-Pro-N-amidinopiperidine structure. Please also note that this particular structure is

characterized by another crystal form with different packing compared to all other structures. Furthermore, data were collected at another beamline, which could explain the significantly lower resolution limit. We currently refrain from relaxing this criterion in our working group in order to keep consistency with former work. For the 5MNN and 5MNG structures we also wanted to avoid a possible criticism that we claim to have ultra-high resolution structures, while we might have questionable statistics for these. With the resolutions reported and statistics presented, we are comfortable to claim ultra-high resolution. We have included the missing CC1/2 statistics in the crystallographic tables of the revised Supporting Information.

3. Isotopes are not normally considered to be distinguishable by X-ray crystal structure analysis. However, it has been suggested that the difference in the vibrational behavior of deuterium and protium could provide a way to distinguish these two isotopic species by X-ray diffraction. Could you confirm it based on your experimental data?

We find this aspect very interesting for crystallographers, however, feel that this is way out of scope of our article and far too specialized information for publication in *Nature Communications*. However, we are willing to provide all the required information and support to interested readers and crystallographers to further investigate the potential of X-ray crystallography to differentiate deuterium and hydrogen.

4. PROCHECK is a bit outdated program for checking the stereochemical quality of a protein structure. In order to conform to “current standards” the stereochemical quality of a protein should be validated using Molprobity program.

We agree with the referee that PROCHECK is a bit outdated by now. Nevertheless, we still use the PROCHECK statistics for those protein structures which we have previously determined with other ligands and published, in order to maintain comparability and consistency to already published quality criteria. For *de novo* structures we usually use MOLPROBITY. The interested reader may find the MOLPROBITY statistics in the PDB where the full validation report is provided for our structures. In addition, we provide the MOLPROBITY statistics in addition to the PROCHECK statistics in the revised version of the crystallographic tables.

A few minor comments.

Page 3. The authors state that “X-ray crystallography is unable to detect hydrogen atoms reliably”. Although the next sentence provides a more detailed description concerning the usual resolution range achieved by that experimental technique, the statement it is not fully correct. Certainly, detection of hydrogens depends on several parameters, e.g.: the quality of crystal, the highest resolution limit, completeness of diffraction data, ADP, disorder/occupancy, etc. Certainly, not all hydrogen atoms can be detected based on the electron density map. However, X-ray crystallography has been successfully used to study enzymes with bound substrates/products at true atomic resolution. Although not all positions of hydrogen atoms could be assigned unambiguously on the basis of the electron-density maps, the protonation state of some, important for the mechanism, groups could be deduced.

We agree with the referee, the detection of hydrogen atoms depends on several parameters. Also in some ultra-high resolution structures hydrogen atoms could be assigned at protein residues and even at some water molecules as for example in BPTI and in our newly reported ultra-high resolution trypsin X-ray structures, but this is rather the exception than the rule. Furthermore, experimentally determined hydrogen positions from X-ray experiments suffer from accuracy due to libration effects

and the fact that the hydrogen electron is not localized at the nucleus. Most often it is the important hydrogen atoms that evade detection with X-rays, whereas in neutron diffraction experiments detection of hydrogen atoms is much more favorable. Nevertheless, we agree with the reviewer that our statement was too absolute and rephrased it more carefully: *"X-ray crystallography can hardly detect hydrogen atoms reliably unless ultra-high resolution data had been collected which may allow detection of the structurally most defined H-atoms"*

Page 6. "certain protein functionalities" sounds a bit odd to me. It could be replaced by "certain protein functional groups such as hydroxyl moiety"

We thank the reviewer for the advice and applied the suggested replacement.

Page 10. "two scattered configurations" could be described as "two distinguished configurations".

We used the proposed phrase in the revised manuscript.

Page 10. "Apart from this interaction, it appears difficult for W1 to find an additional partner for an optimal interaction" sounds personal. Describing it as the lack of additional interaction partners is more neutral.

We agree with the reviewer that such a change results in a scientifically more appealing sentence and therefore rephrased accordingly.

Page 11. The description "this water experiences enhanced residual mobility, it clearly sacrifices rotational degrees of freedom upon ligand accommodation" is not wrong, just a bit personalizing W1 water molecule. :-)

Also in this case, we rephrased to make the statement less "personal".

Reviewer #4 (Remarks to the Author):

Schiebel et al investigates the role of water molecules in the S1 binding pocket in the apo and ligand bound forms of bovine trypsin using X-ray and neutron crystallography. The data appears to be high-resolution and very good quality. I have some concerns about the derived crystallographic models and the conclusions of this study.

Methods:

Crystallography:

The flow of crystallographic models and Rfree flags of reflection have to be better described. Where the starting structure of trypsin was coming from? How were the Rfree flags generated and are they related between the different structures? In most of the structures the gap between Rfree and R is not that great (<1%), suggesting that the Rfree sets were de novo generated and a limited rounds of (automated) refinement were performed. At the same time there is a great variation in Rfree between closely related structures suggesting different origins of Rfree sets (11% vs 17%). For very high resolution X-ray structures allocating 5% Rfree flags is not motivated on statistical grounds (2000-3000 reflections are typically sufficient) and weakens the experimental evidence. Since the paper is focusing very much on the S1 site it is unclear how much scrutiny the rest of the structure

received. Could the evidence substantially alter the crystallographic models during automated refinement?

We are grateful for the referee's comment. Indeed, we missed to mention in the experimental section that the initial phases were obtained by molecular replacement and thus extended our description of the flow of crystallographic models and Rfree flags in the revised manuscript. Furthermore, we would like to point out that we wanted to avoid a full repetition of our refinement strategy that has been fully described in our recent report (Angew. Chem. Int. Ed. Engl. **2017**, *56*, 4887-4890) and to which we are referring right in the beginning of the experimental section in the main manuscript when writing "*All crystallographic experiments and structure determinations were performed following our previously reported strategy.*¹⁴". As a search model the coordinates of PDB entry 4I8H were used. The Rfree flags were independently generated for each data set. We did not copy Rfree sets from one data set to the next due to variations in resolution and thus the total number of reflections. Therefore, as the referee noted correctly, the size of the Rfree sets differs between datasets. In addition, we performed molecular replacement including rigid-body refinement for each structure followed by simulated annealing. In particular the simulated annealing step, which is widely known to remove model bias, has been intentionally performed to ensure that the Rfree is not biased from the search model toward the refinement. The referee noted, that in many cases the difference between R and Rfree is small (<1%). This is to be expected for ultra-high resolution data sets. Rfree serves as a measure to detect over-fitting, a common problem for low resolution data sets with poor data/parameter ratio. For ultra-high resolution data sets such as the ones discussed here, data/parameter ratio is not a problem and we obtain ratios that are common in small-molecule crystallography. There, Rfree only plays a minor role. Once we have lower resolution, as is the case for the 1.34 Å structure of 5MNO, the Rfree raises to 17% with a slightly bigger gap between R and Rfree of 3.4% as one would expect. This lower resolution structure is actually the only structure that has an Rfree that is significantly above the others, which is explained by the much lower resolutions. All other X-ray and XN structures have Rfree values (for the X-ray part) quite close to each other between 10.1 and 12.5% (when omitting the 5MNE structure with the next-lowest resolution of 1.01 Å, the match is actually very good with a range of 10.1 to 11.0% indicating high consistency between those structures of comparable resolution). Moreover, we would like to highlight that refinement was very carefully performed by experienced crystallographers so that the overall number of refinement cycles was quite large further reducing potential model bias. For instance, the finalization of our three XN structures required on average 30 rounds of model building and refinement (each with five macrocycles in PHENIX). To prove our above statements regarding the small R-Rfree gap, please find below (at the end of the paragraph) the distributions of the R-Rfree gap for structures currently deposited in the PDB (PHENIX was used in command-line mode to calculate these statistics by the given commands). It is clearly visible from these histograms that the R-Rfree gap significantly drops with rising resolution. At ultra-high resolution below 1.0 Å as we observed it for most structures in this study, PDB-deposited structures are mostly characterized by a R-Rfree gap of <2%, which is well in agreement with the gaps observed in our cases (e.g. on average 1.3% for the three ultra-high resolution X-ray crystal structures at 100 K; see also the three left columns of Table S5). The referee is also correct, that the size of the Rfree set for the 5MNO and 5MNG structures is a bit high, probably 2% would have been sufficient. We are sorry for this oversight. Nevertheless, this should not be detrimental to refinement since we still have a huge number of reflections available for refinement. If we look for example at the number of reflections gained by the ultra-high resolution (for example, reflections in the last resolution shell for 5MNO are 28611, already 4.5 times the number of Rfree reflections) this is the most important factor contributing to model quality. We are well aware that the deposited structures are a key component of this article and are very confident to have all of them prepared with the required care and scrutiny in all regions of the protein. However, for this publication we believe it is detrimental to focus just on specific aspects

of the wealth of structural information in order to address the broad *Nature Communications* readership. Exemplary for the thoughts we put into these structures we would like to point out that we did not simply add all hydrogen atoms to the models as often done even for neutron structures but rather added only those H- and D-atoms to the model for which experimental evidence in form of sufficient nuclear density (defined threshold of $\pm 2.7 \sigma$ level in the mF_o-DF_c map; as described in the experimental section) was observed after a particular round of refinement although this resulted in a much more time-consuming iterative manual model-building process. As another example, Coot by default displays maps that are filled by F_{calc} for missing reflections. Since neutron diffraction data are in general less complete compared to X-ray diffraction data, we considered this to be a potential source of model error/bias and therefore changed the default for neutron and XN structures to unfilled maps to enable a hopefully minimally biased model-building process.

*Histogram of Rfree-Rwork for all model in PDB at resolution **0.70-1.00 A**:*

0.001 - 0.006	: 12
0.006 - 0.012	: 50
0.012 - 0.017	: 61
0.017 - 0.022	: 90
0.022 - 0.027	: 48
0.027 - 0.033	: 22
0.033 - 0.038	: 8
0.038 - 0.043	: 4
0.043 - 0.049	: 1
0.049 - 0.054	: 5

Number of structures considered: 301

*Histogram of Rfree-Rwork for all model in PDB at resolution **0.80-1.20 A**:*

0.001 - 0.010	: 80
0.010 - 0.019	: 368
0.019 - 0.028	: 485
0.028 - 0.037	: 262
0.037 - 0.045	: 96
0.045 - 0.054	: 40
0.054 - 0.063	: 14
0.063 - 0.072	: 3
0.072 - 0.081	: 1
0.081 - 0.090	: 1

Number of structures considered: 1350

*Histogram of Rfree-Rwork for all model in PDB at resolution **1.80-2.20 A**:*

0.000 - 0.010	: 130
0.010 - 0.020	: 875
0.020 - 0.030	: 3023
0.030 - 0.040	: 5528
0.040 - 0.050	: 5357
0.050 - 0.060	: 3012
0.060 - 0.070	: 1256
0.070 - 0.080	: 449
0.080 - 0.090	: 173
0.090 - 0.100	: 70

Number of structures considered: 19873

Histogram of Rfree-Rwork for all model in PDB at resolution **2.80-3.20 Å**:

0.001 - 0.011	: 62
0.011 - 0.021	: 211
0.021 - 0.031	: 470
0.031 - 0.041	: 791
0.041 - 0.050	: 974
0.050 - 0.060	: 840
0.060 - 0.070	: 502
0.070 - 0.080	: 265
0.080 - 0.090	: 131
0.090 - 0.100	: 67

Number of structures considered: 4313

I am reasonably sure that the S1 pocket was analyzed carefully, but there are other, arguably more important, regions such as the catalytic triad where the position, presence or absence of a hydrogen/deuterium has very important implications. In ultra-high resolution structures there is substantial evidence for disorder. It is necessary to describe how these were modelled, how many residues (and which atoms) were treated as alternative conformers in each structure. Figures showing the model and electron density in these regions are also necessary to validate the model building and refinement strategy.

Clearly, the catalytic triad is very important and the neutron diffraction method is ideally suited to investigate the protonation of this important regions and might reveal insights into the catalytic mechanism. However, the present study was not designed in a way to be optimally suited to analyze the catalytic mechanism but to learn more about ligand recognition in the S1 pocket. If our main goal would have been the investigation of the catalytic triad, we should have used a substrate and a transition-state analog for co-crystallization to get hold of meaningful structural states. In fact, such an analysis (at least with a transition-state analog) has been previously performed using neutron diffraction and this is the second reason why we did not intend to analyze the catalytic triad (A. A. Kossiakoff *et. al.*, *Nature* **1980**, 288, 414-416). Also retrospectively, the analysis of the catalytic triad, which has in fact been performed, did not turn out to add a lot of new information to what was known before. This was the basis for our final decision not to include such an analysis in the manuscript because it would distract the reader from the main message of our article, which is centered on protein-ligand binding, and the role of water in this process. Furthermore, there are many additional interesting findings we made in our XN structures, which we also do not describe because this would fill the manuscript with too much detailed information, which might discourage readers especially when not from within the field. Please also note that all our structures and the associated diffraction data have been deposited in the PDB so that all the information not directly given in our manuscript is still available to readers interested in one specific region of the protein such as the catalytic triad. Nevertheless, we agree with the reviewer that some description of the catalytic triad is useful for the reader due to the close proximity to the S1 pocket. **We therefore added the most important information in the main text (Experimental Section since the question of the reviewer was about the model building and refinement strategy)** with a reference to further information that can be found in the supplement (new Fig. S9). The decision that some information will end up in the supplement was again made on the basis of the above described reasons (e.g. potential distraction of the reader from the main topic of the manuscript, which is the ligand-binding process).

MD simulation:

I appreciate the efforts of the authors to manually curate the hydrogen positions guided by the neutron diffraction data. I think it is reasonable to fill missing positions for non-titrable hydrogens by reasonable prior knowledge, but for titratable hydrogens the experimental evidence must take precedence. Motivation like we protonated the epsilon N of His-57, because some continuum electrostatics method determined so in my view is not acceptable. Otherwise what is the point of doing neutron diffraction experiments? In my opinion, the protonation state of His-57 at pH 7.5 deserve better analysis than hiding it in the supplementary methods section. If for no other reason, because of its close proximity to the S1 binding site.

We absolutely agree with the referee. We always used the hydrogen positions revealed via neutron diffraction experiments for the assignment of protonation states if the nuclear density was sufficient to be indicative of the location of a particular H or D-atom. Only if this was not the case, the hydrogen location suggested by H++ has been used in order to derive a complete and meaningful MD starting structure. In our revised manuscript, we included all methods in the main text in order to more clearly communicate the applied techniques and approaches to the reader. In the revised section describing the above mentioned matter we tried to more clearly express that H++ was only used where neutron data did not give unambiguous H and D-positions by an addition in brackets: "*Hydrogens of this model were placed according to our neutron data or, if not visible in the neutron structures (and only then), as predicted by H++.*". We apologize if we did not state clearly enough that the double protonation of His-57 for the simulation in fact was mainly guided by our neutron structures and only further supported by the H++ simulations. We rephrased this section in order to more clearly convey this message to the reader. By the way, ITC results from a previous study also clearly underlines that His-57 is protonated at physiological pH (J. Mol. Biol. **2007**, 367, 1347–1356). We understand the opinion of the referee concerning His-57. As explained above, we refrained from an analysis of the catalytic triad including His-57 for mainly three reasons. Firstly, our structures mainly confirm previous findings, secondly the study design was not optimized for analysis of the catalytic mechanism of trypsin but for the analysis of the interaction with inhibitors and finally, we believe that an additional focus on the catalytic residues will distract the reader from the main conclusions and purpose of this manuscripts. Nevertheless, we think the referee is right that some description of the catalytic triad is required and therefore added the most important findings in the main text with some additions in the SI (see above).

Conclusion:

It is safe to say that a conclusion with more than 1000 words is excessive and at times it reads like pure discussion/speculation. For example uncharged inhibitors, regulatory sodium ions and comparison of position 192 in trypsin and thrombin are only mentioned in the conclusion without any support in previous sections. I suggest the authors condense their main conclusion into focused points like in their earlier *Angewandte Chemie* paper.

We agree with the reviewer and re-structured the manuscript (partitioning in Results, Discussion, and Conclusion sections). The now much shorter Conclusion section tries to emphasize the evidence and impact of our study for the field of drug design and serine proteases in particular.

Minor points:

Introduction

"Unfortunately, not much experimental data have been collected deciphering how water molecules act exactly during protein-ligand complex formation."

Bulk and interface water was successfully studied with far-IR and optical Kerr effect spectroscopy, I suggest the authors extend their literature study, because experimental data is available.

The referee is correct that there are some experimental data available (e.g. Turton et. al., *Nature Communications* **2014**, 5, 3999). However, our sentence also did not state that we believe there are no experimental data available. There is just not much of it. Nevertheless, to make our statement more transparent, we modified it and, more importantly, extended our literature research and included references to important work in this particular field. Moreover, we would like to note that our sentence particularly refers to the role of water in the ligand recognition process and not so much to the protein-water interaction in general that has been studied to a much greater level of detail, especially by the techniques mentioned by the reviewer.

“The situation improves significantly once the resolution gets better than 2.0 Å with distinct water orientations discernible at 1.5 Å or better.⁷”

Neutron diffraction is also an averaging technique and thus affected by local disorder even at very high resolution. The high-resolution structures presented here do not have 100% detection of hydrogen positions.

We agree with the referee that high resolution will not result in detection of 100% of the hydrogens but we think that we also do not claim this in the above sentence. In order to make this more clear, we rephrased it to "*with the orientation of many waters discernible...*". Also, we would like to mention that we intentionally included the statistics for the number / percentage of protein, water and ligand hydrogens visible in the density of each of our structures (as described in the methods section, only these hydrogens / deuteria with sufficient experimental evidence were modeled). We admit that this information might have been a little hidden in the Supporting Information file and we therefore added a reference to these statistics at a prominent position in the first paragraph of the Discussion. From these statistics one can see that the rephrased above sentence now is clearly correct. Table S7, for instance, shows that 90-91% of the protein hydrogen atoms and 68-75% of the water hydrogen atoms could be identified in our XN structures.

Page 17 “Furthermore, the chloro-substituted ligands establish an electrostatic and thus enthalpically favored chloro-aromatic contact”

London dispersion forces are better explanation for the stabilization of chloro-aromatic contact.

According to Bissantz *et. al.* (*J. Med. Chem.* **2010**, 53, 5061-5084), the most accurate description of the interactions of chloro-substituted ligands are indeed London dispersion forces but also a halogen bond with a backbone CO group. We therefore added the above mentioned reference and rephrased as follows: "*Furthermore, chloro-substituted ligands establish van der Waals and halogen bonding interactions that are likely enthalpically favorable.*³¹"

Reviewers' comments:

Reviewer #1 (Remarks to the Author):

The authors made significant modifications in the manuscript, and paid great attention to provide detailed answers and comments to the points raised in the first round of review.

Regarding the new version of the manuscript, I am in favour of publication of this work, upon slight modifications detailed below.

p3: Based on their unique properties.

p6: fluorescence-based inhibition assay

p9: You mention the comparison of residence times between surface exposed Asp/Tyr and Asp189 / Tyr 228 but do not draw any conclusion.

p9: it is remarkable that the water molecules characteristics ...

p9: "electrostatics of the binding pocket" seems unclear. Maybe use electrostatic surface, or electrostatic properties of the binding pocket.

p9: This sentence is not clear to me: "This difference is even more important.... exchange rate". I am not sure about the message. Could you clarify please.

Reviewer #2 (Remarks to the Author):

The authors have properly addressed the questions raised. I do believe that the jagged profile found in Fig. 4 might be connected with some numerical issues (I do not see how 6 minima could be found in such a short angle range...) but this is a very minor issue.

I support the publication of the manuscript as is (if the authors would care to review the computed profiles, I would be glad if they would do it, but will not demand it).

Reviewer #3 (Remarks to the Author):

The authors have almost satisfactorily responded to most of my questions and made the necessary changes to the manuscript. However, some information provided is not correct and a few important points have to be clarified or fixed before we can proceed and a positive action can be taken.

The newly introduced sentence contains most likely a typo:

1. An intermediate model of the PHENIX refinement has been used as a starting point for the SHELXL refinement. Hydrogens have not been included in the model. Enhanced rigid-body restraints have been applied during refinement (RIGU command).

RIGU command turns on enhanced rigid bond restraints (not rigid body).

2. Using the BLOC instruction does not provide the estimation of required standard uncertainties in a reliable way. From the provided description it is not really clear that only inversion of the full normal matrix, or of large matrix blocks, provides the reliable estimation of the precision of individual parameters if the following prerequisites have been met:

A) The atomic model has been refined using SHELX to convergence. The authors stated that "An intermediate model of the PHENIX refinement has been used as a starting point for the SHELXL refinement" although the final model should have been used instead.

- B) Upon refinement convergence (using SHELX program), the last SHELX refinement should be performed against all data (including Rfree set).
- C) A single full matrix cycle (L.S. 1) using a DAMP 0 0 instruction should be performed while keeping anisotropic displacement parameters fixed (BLOC 1). BLOC 1 instruction reduces size of the matrix and makes the inversion process more stable.
- D) The INS file for SHELX should not contain any instructions defining restraints. These lines should be either removed or commented with REM, e.g.: lines beginning with RIGU, SIMU, DELU, ISOR, etc.

These four conditions must be met if the realistic standard deviations on geometrical parameters are required. However the description provided by the authors questions if the published uncertainties are realistic and unbiased. In my first report I have asked the authors to provide more details about the procedure used because the described protocol was not complete.

I would suggest that the authors provide a more detailed protocol they used in Supplementary Notes and if not all conditions have been met, the standard deviations on geometrical parameters will be newly estimated.

Reviewer #1 (Remarks to the Author):

The authors made significant modifications in the manuscript, and paid great attention to provide detailed answers and comments to the points raised in the first round of review.

Regarding the new version of the manuscript, I am in favour of publication of this work, upon slight modifications detailed below.

p3: Based on their unique properties.

This linguistic error has been corrected.

p6: fluorescence-based inhibition assay

We made this revision and agree that it will help the reader to better understand our approach.

p9: You mention the comparison of residence times between surface exposed Asp/Tyr and Asp189 / Tyr 228 but do not draw any conclusion.

The revised version of the manuscript now contains more details regarding this comparison. Also, a reference to a work from Makarov *et al.* is made. This reference supports our hypothesis about the dynamical behavior of the water molecules adjacent to the side chain of Asp189. Presumably, those water molecules are influenced strongly by the shape of the binding pocket and its electrostatic properties, which enhance the interactions arising from the electrostatic contributions and hydrogen bonding abilities of Asp189.

p9: it is remarkable that the water molecules characteristics ...

The sentence has been removed in the revised version of the manuscript since it overlapped with a more elaborate conclusion regarding the comparison of residence times between Asp189/Tyr228 and surface exposed Asp/Tyr.

p9: "electrostatics of the binding pocket" seems unclear. Maybe use electrostatic surface, or electrostatic properties of the binding pocket.

We agree and completely removed this phrase in the revised version of the manuscript for the reason given above.

p9: This sentence is not clear to me: "This difference is even more important.... exchange rate". I am not sure about the message. Could you clarify please.

We rephrased this sentence and provided additional information in the revised version of the manuscript. Our aim was to provide a hypothesis about the functional role of the observed differences in residence times between solvent sites W1 and W2/W3. The fast exchange rate observed for water molecules at W1 are indicative of a low unbinding barrier for those water molecules and as such also can be used as unbinding pathway for other water molecules in the S1 pocket. However, this requires the exchange of water molecules between solvent site W1 and the remaining solvent sites in the unbound or pre-binding states, which then is followed by an unbinding of these water molecules. We do not want to extend the current study by more in-depth investigations of the water rearrangement mechanism in the binding pocket, since it is also part of an ongoing study in our group and beyond the scope of the current work. Nevertheless, the current version of the manuscript contains more details regarding the functional role of these residence times, which hopefully provides a clearer conclusion.

Reviewer #2 (Remarks to the Author):

The authors have properly addressed the questions raised. I do believe that the jagged profile found in Fig. 4 might be connected with some numerical issues (I do not see how 6 minima could be found in such a short angle range...) but this is a very minor issue.

I support the publication of the manuscript as is (if the authors would care to review the computed profiles, I would be glad if they would do it, but will not demand it).

All QM and QM/MM calculations in the manuscript were performed using a consistent CP2K-based approach. With this approach, a smooth energy profile resulted for *N*-amidinopiperidine alone. On the basis of the reviewer's concern, we reviewed the jagged energy profile of the complex under the applied computational conditions and we suppose that the fluctuations were observed due to a strong sensitivity to small structural adaptations in the Asp:*N*-amidinopiperidine complex (e.g. orientation between the Asp and guanidino groups). Nevertheless, if we consider the mean trend of the computed energies, no significant stabilization of neither the planar nor the pyramidal geometry is indicated for the complex. This is well in line with our experimental observation of both geometries side-by-side in the crystal structure and with our QM/MM calculations that support the existence of the pyramidal form in the trypsin-bound state and the planar arrangement outside the protein. Moreover, the resulting conclusion is further supported by our CSD searches. This all is consistent with the calculated energy profiles. Since, however, the latter calculations are not in the focus of the present study and only assist to explain the experimental observation, we finally decided to remove

the energy profiles and all related methodological descriptions from the manuscript, which also makes the manuscript more concise. In order to keep methodological consistency throughout the manuscript (only CP2K calculations) and due to the minor importance of these calculations that are anyhow in the gas phase and not in solution as the much more important QM/MM calculations, we would like to refrain from starting alternative and more sophisticated QM calculations for the prediction of energy profiles. However, we kept the results from the (converged) unrestrained geometry optimizations in the manuscript, which suffice to show that the pyramidal *N*-amidinopiperdine form is only stable in the presence of the aspartate counter-ion. The corresponding orbital representations, which were originally combined with the energy profiles from the restrained QM calculations can now be found in the supplement.

Reviewer #3 (Remarks to the Author):

The authors have almost satisfactorily responded to most of my questions and made the necessary changes to the manuscript. However, some information provided is not correct and a few important points have to be clarified or fixed before we can proceed and a positive action can be taken.

The newly introduced sentence contains most likely a typo:

1. An intermediate model of the PHENIX refinement has been used as a starting point for the SHELXL refinement. Hydrogens have not been included in the model. Enhanced rigid-body restraints have been applied during refinement (RIGU command).

RIGU command turns on enhanced rigid bond restraints (not rigid body).

We thank the reviewer for his attention and apologize for this error. We were misled by a remark when generating the .ins file with pdb2ins. Here it states:

RIGU !Apply enhanced rigid body restraints

Of course, that should have been avoided by properly reading the program manual. The error in the remark generation has been reported to the program author. Also it has been corrected in the manuscript together with a more detailed description of the refinement process.

2. Using the BLOC instruction does not provide the estimation of required standard uncertainties in a reliable way. From the provided description it is not really clear that only inversion of the full normal matrix, or of large matrix blocks, provides the reliable estimation of the precision of individual parameters if the following prerequisites have been met:

A) The atomic model has been refined using SHELX to convergence. The authors stated that "An intermediate model of the PHENIX refinement has been used as a starting point for the SHELXL refinement" although the final model should have been used instead.

B) Upon refinement convergence (using SHELX program), the last SHELX refinement should be performed against all data (including Rfree set).

C) A single full matrix cycle (L.S. 1) using a DAMP 0 0 instruction should be performed while keeping anisotropic displacement parameters fixed (BLOC 1). BLOC 1 instruction reduces size of the matrix and makes the inversion process more stable.

D) The INS file for SHELX should not contain any instructions defining restraints. These lines should be either removed or commented with REM, e.g.: lines beginning with RIGU, SIMU, DELU, ISOR, etc.

These four conditions must be met if the realistic standard deviations on geometrical parameters are required. However the description provided by the authors questions if the published uncertainties are realistic and unbiased. In my first report I have asked the authors to provide more details about the procedure used because the described protocol was not complete.

I would suggest that the authors provide a more detailed protocol they used in Supplementary Notes and if not all conditions have been met, the standard deviations on geometrical parameters will be newly estimated.

We are grateful to the reviewer for the advice in performing proper refinement for estimation of standard deviations. We carefully followed the individual steps outlined above by the referee in our refinement. While we had performed a very similar refinement, we had not used the fully refined structure from Phenix, which we have done now by using the PDB deposited structure (PDB code 5MNN). We have also included a much more detailed refinement protocol in the experimental section. Importantly, the application of this modified refinement procedure did result in only minor changes in the reported values and therefore did not conflict with any of the manuscript's conclusions. All reported values were updated in the text. Also, Fig. 3C and F have been updated on the basis of the results of the new refinement procedure.